# Investigating grey matter volumetric trajectories through the lifespan at the individual level

Runye Shi [1,32], Shitong Xiang [2,3,32], Tianye Jia [2,3,4,5,6,32], Trevor W. Robbins [2,3,7], Jujiao Kang [2,3], Tobias Banaschewski [8], Gareth J. Barker [9], Arun L. W. Bokde [10], Sylvane Desrivières [4], Herta Flor [11,12], Antoine Grigis [13], Hugh Garavan [14], Penny Gowland [15], Andreas Heinz [16], Rüdiger Brühl [17], Jean-Luc Martinot [18], Marie-Laure Paillère Martinot [18,19], Eric Artiges [18,20], Frauke Nees [8,11,21], Dimitri Papadopoulos Orfanos [13], Tomáš Paus [22,23], Luise Poustka [24], Sarah Hohmann [8], Sabina Millenet [8], Juliane H. Fröhner [25], Michael N. Smolka [25], Nilakshi Vaidya [26], Henrik Walter [16], Robert Whelan [27], Gunter Schumann [5,26], Xiaolei Lin [1,28] ✉, Barbara J. Sahakian [2,3,7] ✉, Jianfeng Feng [1,2,3,29,30,31] ✉ & IMAGEN Consortium*

Adolescents exhibit remarkable heterogeneity in the structural architecture of brain development. However, due to limited large-scale longitudinal neuroimaging studies, existing research has largely focused on population averages, and the neurobiological basis underlying individual heterogeneity remains poorly understood. Here we identify, using the IMAGEN adolescent cohort followed up over 9 years (14–23 y), three groups of adolescents characterized by distinct developmental patterns of whole-brain gray matter volume (GMV). Group 1 show continuously decreasing GMV associated with higher neuro-cognitive performances than the other two groups during adolescence. Group 2 exhibit a slower rate of GMV decrease and lower neurocognitive performances compared with Group 1, which was associated with epigenetic differences and greater environmental burden. Group 3 show increasing GMV and lower baseline neurocognitive performances due to a genetic variation. Using the UK Biobank, we show these differences may be attenuated in mid-to-late adulthood. Our study reveals clusters of adolescent neurodevelopment based on GMV and the potential long-term impact.

Adolescence is a critical and active period for brain reconstruction and maturation, with regional changes of synaptic morphology, dendritic arborization, cortical cell firing, and changes in neurochemical receptor affinity[1–3]. The risk for many neuropsychiatric disorders increases during this period, including conduct disorder, mood disorder, and schizophrenia[4–6]. Structural neurodevelopment during adolescence is important for enhanced cognitive abilities and mental well-being persisting into adulthood[7–11]. Population-based studies have shown that adolescents exhibit remarkable heterogeneity in terms of structural neurodevelopment[12–14], but the neurobiological basis of the

---

A full list of affiliations appears at the end of the paper. *A list of authors and their affiliations appears at the end of the paper.
✉ e-mail: xiaoleilin@fudan.edu.cn; bjs1001@cam.ac.uk; jianfeng64@gmail.com

heterogeneity remains poorly understood. Most efforts have been devoted to study the functional circuitry and structural composition of the brain and their associations with mental health disorders at the population level[1,15–18]. These pioneering studies have leveraged large population cohorts and refined our understanding of the adolescent brain. However, associations between behavioral patterns and trajectories of brain development vary at the individual level and understanding the sources of variation remains imperative in the arena of public health and precision medicine[9,13,19,20].

Large-scale longitudinal neuroimaging studies have enabled delineation of the dynamic changes of individual brain morphology, by clustering adolescents according to their developmental trajectories of neuroimaging-derived phenotypes. Similar approaches have yielded associations between atypical brain structure and neuroanatomical variation across neuropsychiatric disorders[21]. Neuroimaging biomarkers offered tremendous versatility to determine the neuropathological mechanisms of neurodegenerative and mental illnesses[22–24], but have yet not been fully utilized for neurodevelopment. Structural magnetic resonance imaging (sMRI) provides noninvasive measures of imaging-derived phenotypes, among which the developmental courses of gray matter volume (GMV) were shown to be strongly associated with myelinogenesis and synaptic plasticity during adolescence[25–28]. Collectively, this raises the possibility of identifying distinct clusters of dynamic brain structure according to the growth trajectories of whole-brain GMV architecture.

In this study, we aim to investigate the individual heterogeneity of adolescent brain development, potential genetic, epigenetic, and environmental factors that could contribute to the heterogeneity, and possible long-term impacts of the heterogeneous brain developmental patterns on the biological and social wellbeing later in life. To accomplish these goals, we employ a data-driven approach to cluster adolescents into groups with distinct whole-brain GMV developmental patterns using longitudinal neuroimaging data from the IMAGEN cohort that spanned the entire period of adolescence and early adulthood (schematic workflow in Fig. 1a). Both genome-wide and epigenome-wide association studies (EWAS) are conducted to dissect the genetic and epigenetic variations associated with each cluster. A limitation of our study is that due to limited sample size and to avoid confounding effects of ethnicity in this small sample, our study only included participants that self-reported as white. It is worth noting that, in order to extend the investigation from adolescence to late childhood and mid-to-late adulthood, we bridged IMAGEN to Adolescent Brain Cognitive Development study (ABCD) and UK Biobank (UKB) through different mapping approaches assuming population homogeneity. Specifically, longitudinal brain changes were mapped to baseline neuroimaging phenotypes in IMAGEN, which were further used to evaluate the associations between cross-sectional brain measures and population cluster in ABCD, assuming comparable linear changes from late childhood to adolescence for each structural brain measure. Genomic and neuroimaging data in ABCD allowed us to identify potential genetic variations associated with particular population clusters. Finally, genomic, neuroimaging, and other related phenotypes in UKB allowed us to investigate the long-term impact of genetic-proxied neurodevelopment.

## Results

### Developmental trajectories of whole-brain GMV during adolescence define three clusters

We began by estimating the longitudinal trajectories of GMV in 44 brain regions of interest (ROIs) (34 cortical and 10 subcortical ROIs) that spanned the whole brain of each adolescent across baseline (at age 14 y) and two follow-up scans (at age 19 y and 23 y) in the IMAGEN study, adjusting for intracranial volume (ICV), sex, handedness and site (Methods). Individuals showed strong heterogeneity and clustering

patterns in terms of baseline total GMV and GMV developmental trajectories (Supplementary Fig. 1). Next, we reasoned that neurobiologically meaningful clusters could be explained by the developmental patterns in a subset of ROIs. Therefore, we conducted dimension reduction via principal component analysis (PCA) and selected the first 15 principal components (PCs), which explained 80% of the total variation in whole-brain GMV trajectories, in the clustering analysis (Supplementary Table 1 and Supplementary Fig. 2a). The first and second PCs, which accounted for 41% and 6% of the variance, defined two combinations of GMV trajectories over the entire brain that were significantly associated with baseline total GMV (Supplementary Fig. 2b). However, they exhibited different association patterns with items of the Cambridge Gambling Task, where PC1 was significantly associated with delay aversion ($r = 0.07$, $P_{adj} = 0.030$) and risk adjustment ($r = −0.08$, $P_{adj} = 0.020$), and PC2 was significantly associated with deliberation time ($r = 0.1$, $P_{adj} = 0.003$), overall betting (proportion bet) ($r = 0.07$, $P_{adj} = 0.014$) and risk-taking ($r = 0.08$, $P_{adj} = 0.008$) (Supplementary Fig. 2c). These PCs were then used in the multivariate clustering to identify groups of adolescents with distinct neurodevelopmental patterns.

Among 1543 adolescents with at least two sMRI scans, our analyses identified three clusters of structural neurodevelopment ($P_{permutation} < 0.001$) (Supplementary Fig. 3). Group 1 consisted of 711 (46.1%) adolescents, had high baseline total GMV and continuously decreasing GMV at follow-ups, which was consistent with the population GMV developmental trend[21]. Group 2 included 765 (49.6%) adolescents and compared to Group 1, they had lower baseline total GMV, lower peak GMV, and slower rate of GMV decrease. In addition, adolescents in Group 2 are more likely to be older ($Diff = 0.11$ y, $P < 0.001$) and be males ($Diff = 9.5\%$, $P < 0.001$), have parents with lower education attainment ($P = 0.020$ for maternal education; $P = 0.003$ for paternal education) and lower WISCIV full score at age 14 ($Diff = −1.76$, $P = 0.011$). The remaining 67 (4.3%) belonged to Group 3, among whom we observed the lowest baseline total GMV and surprisingly increasing GMV at follow-ups, which was opposite to the population developmental trend (Fig. 1b, Supplementary Figs. 4–6). Compared to Group 1, Adolescents in Group 3 are more likely to have parents with lower education ($P = 0.015$ for maternal education; $P = 0.010$ for paternal education) and lower WISCIV full score at age 14 ($Diff = −9.22$, $P < 0.001$). The full demographic and baseline characteristics for each group were provided in Supplementary Table 2. Since we aim to investigate group-specific brain developmental patterns in late childhood, we further estimated the age-specific GMV growth rate in each ROI from age 5 y to 25 y in each group (Methods) using population neurodevelopmental curve as a reference. Consistently we observed continuously decreasing GMV in Group 1 and Group 2 (with slower rate of GMV decrease in Group 2), and increasing GMV in Group 3 for most ROIs (Fig. 1c), indicating delayed neurodevelopment and brain maturation in Group 3 compared to the other groups, where delayed neurodevelopment was proxied using later peaking time of total GMV.

To understand the neurobiological basis of group heterogeneity, we next tested for differences in whole-brain GMV development among these groups. We observed common delayed GMV development in ROIs spanning the inferior temporal, middle temporal, lateral orbitofrontal, precentral and superior frontal areas in Group 3 (relative to Groups 1/2) (Fig. 1d top and Supplementary Table 3). Group 2 showed lower peak GMV and slower rate of GMV decrease in ROIs spanning superior frontal, caudal middle frontal, rostral middle frontal, precentral, and inferior parietal areas (relative to Group 1) (Fig. 1d bottom and Supplementary Table 3). These are all among the last areas in the brain to mature and had been implicated to play a key role in executive functions. This led us to ask whether variations in structural neurodevelopment could predict the developmental trajectories of neurocognition and risk of neuropsychiatric disorders in these groups.

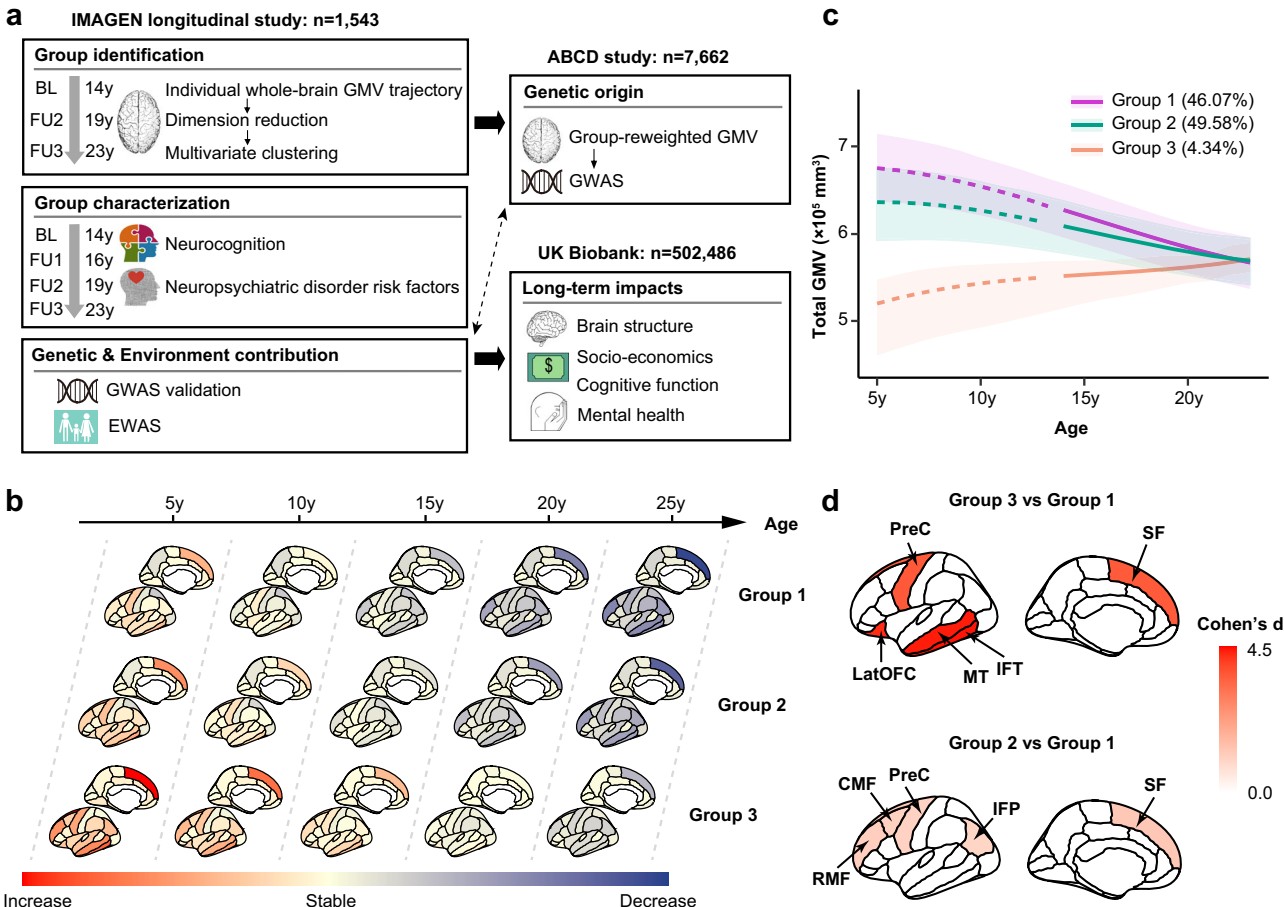

**Fig. 1 | Whole-brain gray matter volume (GMV) developmental patterns define three neurodevelopmental groups. a** Schematic workflow of the analytic methodologies. GMV trajectory in 44 ROIs spanning the whole brain was estimated for adolescents in the IMAGEN study ($n = 1543$). Multivariate clustering was conducted to identify groups with distinct neurodevelopmental patterns, followed by group characterization. Genome-wide association study (GWAS) was conducted in the ABCD study ($n = 7662$) using the proxy phenotype, and epigenome-wide association study (EWAS) was conducted in IMAGEN ($n = 909$). Last, long-term impacts of the polygenic risk for delayed neurodevelopment were investigated among participants in UK Biobank ($n = 502,486$). BL, baseline; FU, follow-up. **b** Whole-brain GMV growth rates (ranging from increase, stable to decrease) at age 5 y, 10 y, 15 y, 20 y and 25 y were estimated for each group, adjusting for sex, imaging site, handedness, and intracranial volume. Group 3 showed delayed GMV development compared to Group 1 and 2. **c** Total GMV developmental trajectories (with 95% confidence bands; the center of the band represents the estimated mean total GMV trajectories within each group) for the three groups (purple for Group 1; green for Group 2; orange for Group 3). These trajectories were estimated adjusting for sex, imaging site, handedness, and intracranial volume. Group 1 and 2 exhibited similar GMV developmental trend, while Group 3 had opposite GMV developmental trend. **d** Top 5 discriminating ROIs with largest t values comparing the GMV trajectories between Group 3 ($n = 67$) and Group 1 ($n = 711$) (top), and between Group 2 ($n = 765$) and Group 1 ($n = 711$) (bottom), adjusting for sex, imaging site, handedness and intra-cranial volume. Two sample two-tailed t-test: Group 3 vs Group1, IFT ($d = 4.43$, $t = 20.13$, $P_{adj} < 0.001$), MT ($d = 4.38$, $t = 20.07$, $P_{adj} < 0.001$), LatOFC ($d = 4.26$, $t = 18.31$, $P_{adj} < 0.001$), PreC ($d = 3.63$, $t = 18.11$, $P_{adj} < 0.001$), SF ($d = 3.61$, $t = 17.92$, $P_{adj} < 0.001$); Group 2 vs Group 1, SF ($d = 1.28$, $t = 24.50$, $P_{adj} < 0.001$), RMF ($d = 1.14$, $t = 21.95$, $P_{adj} < 0.001$), CMF ($d = 1.09$, $t = 20.77$, $P_{adj} < 0.001$), PreC ($d = 1.05$, $t = 20.14$, $P_{adj} < 0.001$), IFP ($d = 1.00$, $t = 19.07$, $P_{adj} < 0.001$). LatOFC lateral orbitofrontal cortex, RMF rostral middle frontal, CMF caudal middle frontal, SF superior frontal, PreC precentral, MT middle temporal, IFT inferior temporal, IFP inferior parietal. Relevant source data were provided in the Source Data file.

## Structural neurodevelopment predicts neurocognition and risk factors for neuropsychiatric disorders

To investigate the association between neurodevelopment and executive functions, we tested for differences of neurocognitive performance among these groups at baseline and at the last follow-up. Full results of these comparisons were provided in Supplementary Table 4. We found that compared to Group 1, Group 3 with delayed neurodevelopment showed worse neurocognitive performance (Spatial Working Memory, Cambridge Gambling Task (CGT) and Stop Signal Task (SST)) at baseline, but most of these items improved over time with brain maturation and became statistically equivalent (two-tailed t-test: $P_{adj} > 0.05$) at the last follow-up (Fig. 2a and Supplementary Fig. 7a). This can be predicted by the structural architecture of GMV development in Group 3, where increasing GMV in the top discriminating ROIs showed positive correlation with improvements of

neurocognition (Supplementary Fig. 8 and Supplementary Table 5). In contrast, Group 2 with slower rate of GMV decrease showed worsened neurocognitive performance (CGT and SST) at the last follow-up relative to baseline (Fig. 2a and Supplementary Fig. 7b), which could be predicted by the negative correlations between the GMV developmental trajectories in the top discriminating ROIs and neurocognition (Supplementary Fig. 8 and Supplementary Table 6).

The delayed brain and neurocognitive development in Group 3 led us to ask whether these adolescents had increased risk for neuropsychiatric disorders. Consistent with the improvements of neurocognition, we observed decreased attention-deficit/hyperactivity disorder (ADHD) symptom. However, in contrast to improved neurocognition, we observed increased depression symptoms in Group 3 (Fig. 2b and Supplementary Table 4). This indicated that although neurocognitive abilities in Group 3 exhibited pronounced

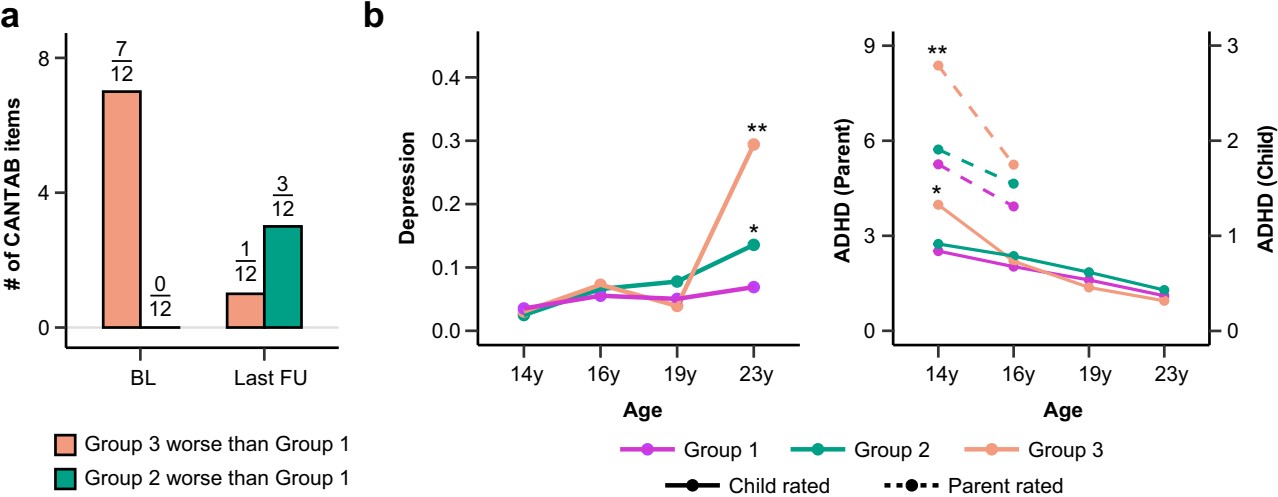

**Fig. 2 | Structural neurodevelopment predicts neurocognition and risk factors for neuropsychiatric disorders. a** Comparison of neurocognitive performances between Group 3 and Group 1 (orange), and between Group 2 and Group 1 (green) at baseline (BL) and the last follow-up (FU). Total number of neurocognitive tests in CANTAB where Group 3 performed worse than Group 1 decreased from 7/12 at BL to 1/12 at the last FU, while the number of tests where Group 2 performed worse than Group 1 increased from 0 to 3/12. Full results with item-specific comparisons among these groups are provided in Supplementary Fig. 7. CANTAB, Cambridge Neuropsychological Test Automated Battery. **b** Longitudinal trajectories of Depression (Left) and ADHD symptoms (Right) among adolescents in three groups (purple for Group 1; green for Group 2; orange for Group 3). Group-specific means at each visit were plotted and * indicated significant differences relative to Group 1 adjusting for sex, handedness, stie and ICV. Baseline mental health score was also adjusted for comparison at the last follow-up. Two-tailed t-tests were conducted at baseline (14 y) and the last follow-up. BH-FDR method was used for multiple correction. Depression, Group 2 vs Group 1 at 14 y ($d = -0.05$, $P_{adj} = 0.256$), Group 2 vs Group 1 at 23 y ($d = 0.13$, *$P_{adj} = 0.023$), Group 3 vs Group 1 at 14 y ($d = -0.01$, $P_{adj} = 0.566$), Group 3 vs Group 1 at 23 y ($d = 0.70$, **$P_{adj} = 0.001$); Parent rated ADHD (dashed line), Group 2 vs Group 1 at 14 y ($d = 0.04$, $P_{adj} = 0.220$), Group 2 vs Group 1 at 16 y ($d = -0.03$, $P_{adj} = 0.574$), Group 3 vs Group 1 at 14 y ($d = 0.34$, **$P_{adj} = 0.004$), Group 3 vs Group 1 at 16 y ($d = 0.01$, $P_{adj} = 0.954$); Child rated ADHD (solid line), Group 2 vs Group 1 at 14 y ($d = -0.03$, $P_{adj} = 0.321$), Group 2 vs Group 1 at 23 y ($d = 0.02$, $P_{adj} = 0.758$), Group 3 vs Group 1 at 14 y ($d = 0.34$, $P_{adj} = $*0.042$), Group 3 vs Group 1 at 23 y ($d = -0.10$, $P_{adj} = 0.579$). ADHD, attention-deficit/hyperactivity disorder. Relevant source data were provided in the Source Data file.

improvement during adolescence, this was not necessarily the case for mental disorder symptoms. Furthermore, consistent with their worsened neurocognitive performances, we observed increased depression symptoms in Group 2 at the last follow-up compared to baseline, (Fig. 2b and Supplementary Table 4). The continuously worsened neurocognition and mental health problems in Group 2 indicated biological, social and mental disadvantages among these adolescents.

Given the slightly different patterns of GMV development for males and females[21], we conducted the analyses stratified by sex following the same workflow. Results of group clustering largely overlapped with the original analyses (Supplementary Table 7). In general, the sex-stratified analyses revealed similar patterns of neurocognition and mental health symptoms among three groups of adolescents. However, differences of neurocognition among these groups were manifested more for risk-taking and impulsive behaviors in males, while for spatial working memory in females (Supplementary Table 8). Besides, increase of the depressive symptoms in Group 2 was only observed in males, and increase of the depressive symptoms in Group 3 was only observed in females.

In addition, we compared the genetic liability to major neurodevelopmental disorders and related traits, including ADHD, autism spectrum disorder (ASD), educational attainment (EA) and intelligence (IQ), by calculating the corresponding polygenic scores (PGS) for each adolescent. Group 3 had higher PGS for ADHD than both Group 1 ($P_{adj} = 0.007$) and Group 2 ($P_{adj} = 0.017$), while Group 2 was not statistically different from Group 1 ($P_{adj} = 0.424$). We did not observe significant differences among the three groups in terms of the PGS of ASD, EA and IQ (Supplementary Table 9). The higher genetic liability of ADHD in Group 3 led us to ask whether genetic variants could explain the delayed neurodevelopment and neurocognitive performances in this group.

## Genetic and epigenetic variations contribute to structural neurodevelopment

To better understand the genetic basis of structural neurodevelopment, we conducted genome-wide association studies (GWAS) for Group 3 versus Groups 1/2 using group-reweighted GMV as the proxy-phenotype among 7662 adolescents in ABCD, since GWAS was underpowered for the IMAGEN study due to limited sample size[29]. Group-reweighted GMV was derived and used as the proxy phenotype because GMV developmental patterns could not be estimated in ABCD due to limited age range. This continuous phenotype represented one's tendency of being in Group 3 relative to Groups 1/2, or in other words, one's propensity of having delayed brain development. Specifically, we began by calculating the ROI-specific weight in discriminating Group 3 (relative to Groups 1/2) in IMAGEN using baseline neuroimaging data adjusting for potential confounders, and applying these weights to corresponding ROIs in ABCD baseline data to obtain the Group3-reweighted GMV, which was then used as the proxy phenotype in the Group 3 GWAS (Methods). The Group3-reweighted GMV showed negative correlation with neurocognition in ABCD, indicating the validity of using Group3-reweighted GMV as appropriate proxy for delayed neurodevelopment (Fig. 3a and Supplementary Table 10). Similarly, Group2-reweighted GMV was calculated and used as the proxy phenotype in the Group 2 GWAS.

One locus showed genome-wide significant effects in the Group 3 GWAS (Fig. 3b and Supplementary Table 11). The lead single-nucleotide polymorphisms (SNP), rs9375442 ($\beta = 0.51$, $P = 9.25 \times 10^{-9}$) on chromosome 6, is an intronic variant located on *CENPW* (Supplementary Fig. 9). *CENPW* is a protein-coding gene involved in the packaging of telomere ends and cell cycle mitotic[30,31], and increased *CENPW* expression in progenitors could lead to decreased cortical volume and cognitive function by altering neurogenesis or increasing apoptosis[32].

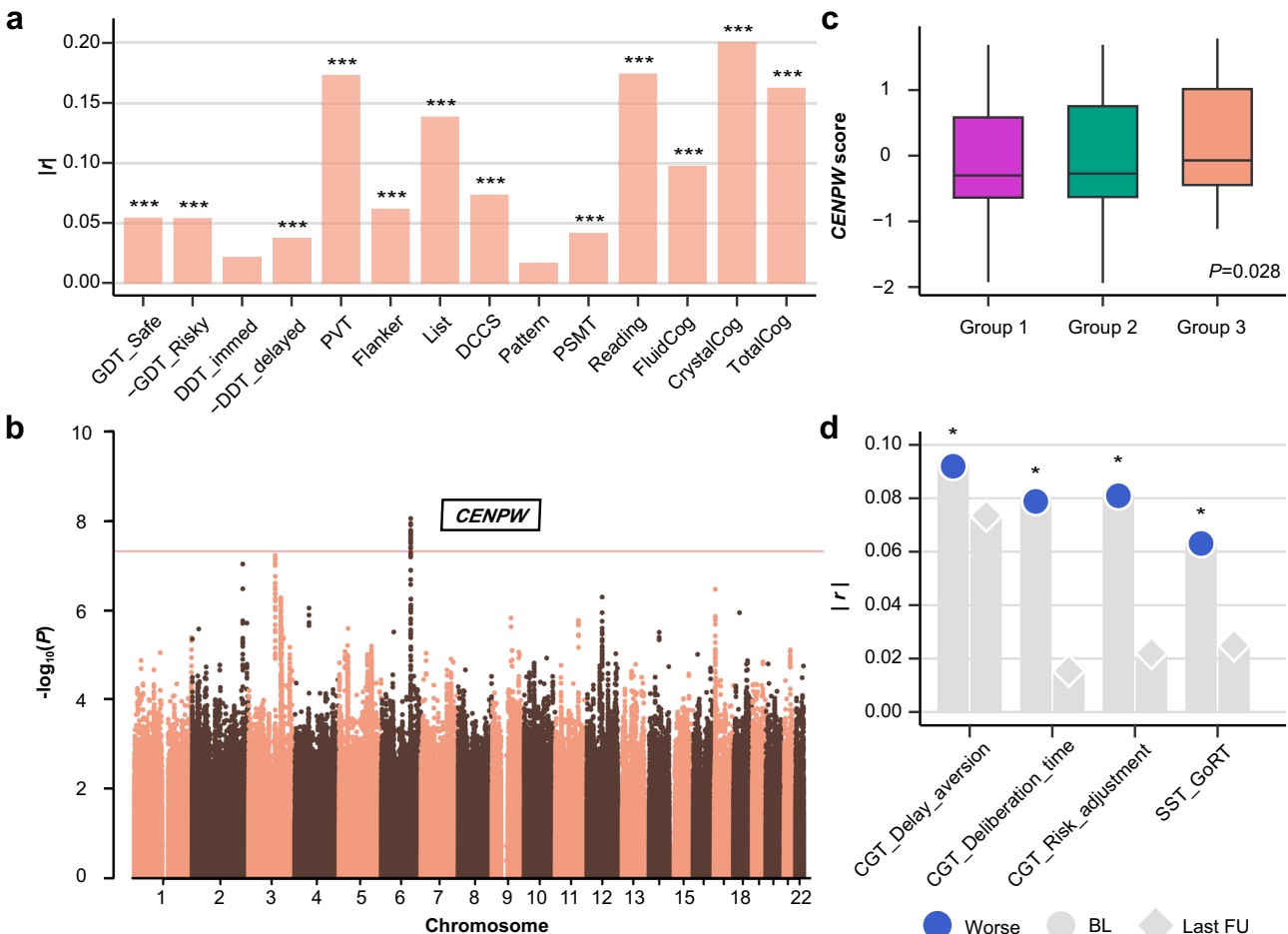

**Fig. 3 | Genome-wide association study (GWAS) identified one significant locus associated with delayed neurodevelopment in Group 3. a** Correlation between Group3-reweighted GMV and neurocognition in ABCD ($n = 11,101$) indicated the validity of using the proxy phenotype for delayed neurodevelopment in the GWAS. One sample two-sided t test was used with FDR for multiple correction. The neurocognition measures and corresponding abbreviations are defined in the Methods and Supplementary Table 10 with exact $p$ values. ***$P < 0.001$. **b** GWAS Manhattan plot for Group3-reweighted GMV in the ABCD population ($n = 7662$). Group3-reweighted GMV was calculated for each adolescent (details in Methods) and used as the proxy phenotype for delayed neurodevelopment. Multiple SNPs on chromosome 6 achieved genome-wide significant effects (two-sided t-test: $P < 5 \times 10^{-8}$), mapped to the intronic region of CENPW. Results from gene-based association analysis (Supplementary Fig. 11) confirmed the significant effect of CENPW on delayed neurodevelopment. Box plot in (**c**) showed that CENPW score of delayed neurodevelopment was higher in Group 3 ($n = 60$) compared to Group 1 and 2 ($n = 1338$) (two-sided t-test: $P = 0.028$). The upper and lower boundaries of each

boxplot represented the first (Q1) and third (Q3) quantiles, respectively. Hence, the box body covered 50% of the central data, with the median marked by a central line. The top/bottom whiskers represented the maximum or minimum, respectively without outliers. **d** indicated that CENPW score of delayed neurodevelopment was negatively correlated with baseline (BL) neurocognitive performance, and became non-significant at the last follow-up (FU). Here, Worse indicated higher CGT Delay aversion score, lower CGT risk adjustment score, longer CGT Deliberation time and SST GoRT. One-sided $P$ values were reported (one sample t test) and BH-FDR method was used for multiple correction within scales. CGT Delay aversion, BL ($r = 0.09$, *$P_{adj} = 0.027$), FU3 ($r = -0.07$, $P_{adj} = 0.239$); CGT Deliberation time, BL ($r = 0.08$, *$P_{adj} = 0.027$), FU3 ($r = 0.02$, $P_{adj} = 0.983$); CGT risk adjustment, BL ($r = -0.08$, *$P_{adj} = 0.027$), FU3 ($r = 0.02$, $P_{adj} = 0.983$); SST GoRT, BL ($r = -0.06$, *$P_{adj} = 0.038$), FU3 ($r = -0.02$, $P_{adj} = 0.472$). CGT, Cambridge Gambling Task; SST GoRT, reaction time for 'Go' trials in Stop Signal Task. **c**, **d** confirmed the relationship between CENPW and delayed neurodevelopment identified in (**b**). Relevant source data were provided in the Source Data file.

Other variants on these genes were reported to be associated with cortical surface area and brain volume[33–36] (Supplementary Fig. 10), general cognitive ability[37–39] and physical growth[40–42]. Gene-based association analysis confirmed the identification of CENPW (Supplementary Fig. 11). Next, we conducted validation of the Group 3 GWAS back in IMAGEN. We began by calculating the PGS for SNPs ($N_{SNP} = 4$) residing in CENPW (CENPW score) and across the whole genome (PGS) that are associated with Group3-reweighted GMV for each adolescent in IMAGEN, tested for the differences of PGS among these groups, and correlated the PGS with neurocognition and behavioral risk factors. Consistent with the Group 3 GWAS, we observed higher CENPW score in Group 3 relative to Groups 1/2 (Fig. 3c) and positive correlations between CENPW score and improvement of neurocognition and conduct problems (Fig. 3d). Similar results were obtained for PGS (Supplementary Fig. 12).

No genome-wide significant SNPs were identified in the Group 2 GWAS (Supplementary Fig. 13). However, the large overlap between the neurodevelopmental patterns and homogeneous genetic liability for neurodevelopmental disorders and related traits (ADHD, ASD, IQ, and EA) in Groups 1/2 led us to reason that the differences of neurocognitive performances between Group 1 and 2 were quantitative (rather than qualitative) and might be influenced by the effects of environmental exposure. This was also supported by the baseline differences in socioeconomic and family factors, such as stressor scores of socioeconomic/housing ($d = 0.30$, $P_{adj} < 0.001$), health ($d = 0.16$, $P_{adj} = 0.014$), relationship/addiction ($d = 0.29$, $P_{adj} < 0.001$) and family affirmation ($d = -0.11$, $P_{adj} = 0.045$) in Group 2 versus Group 1. To test this, we performed EWAS in IMAGEN (Methods) using group label as the phenotype. A significant hypermethylation site cg06064461 ($\beta = 25.40$, $P = 4.24 \times 10^{-8}$) (Fig. 4a) was identified and

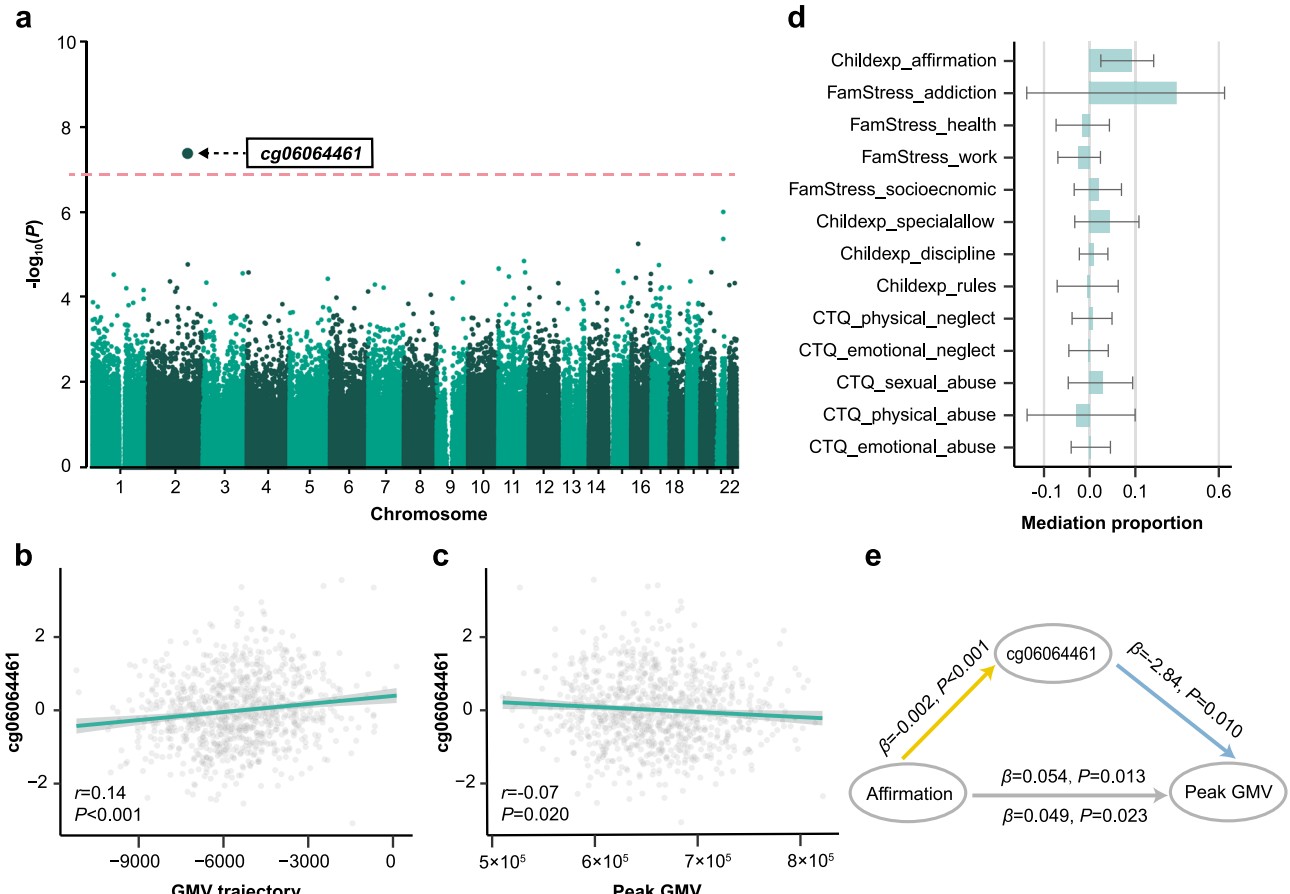

**Fig. 4 | Epigenome-wide association study (EWAS) identified significant signals associated with lowered neurodevelopment in Group 2. a** EWAS Manhattan plot in the IMAGEN population. Group 2 ($n = 463$) (relative to Group 1, $n = 446$) status was used as the phenotype, adjusting for potential confounders. One hypermethylated site cg06064461 achieved genome-wide significant effect (one sample two-sided t-test: $P < 5 \times 10^{-8}$, BH-FDR corrected $P_{adj} < 0.05$) and was mapped to ATF2 and MIR933 on chromosome 2. Validation of EWAS results in IMAGEN ($n = 909$). cg06064461 methylation was positively correlated with total GMV trajectory (**b**; $r = 0.14$, $P = 6.85 \times 10^{-6}$) and negatively correlated with peak gray matter volume (GMV) (**c**; $r = -0.07$, $P = 0.020$), adjusting for potential confounders. The error bands in (**b**, **c**) represent the pointwise 95% confidence intervals of the corresponding estimated correlations. One sample t-test was used. **d** Proportion of the mediation effects through cg06064461 methylation in the environmental exposure - peak GMV pathway, adjusting for potential confounders ($n = 750$ independent samples; the estimates and standard deviation of mediation proportion were estimated using the 1000-iteration bootstrap approach). The bar, also the central of

the error bars, represents the point-wise estimated mediation proportion, while error bars indicate 95% confidence intervals of the estimated mediation proportion. Thus, the left/right whiskers represent the lower bound and upper bound of the confidence interval, respectively. Environmental factors were sorted by P values of the corresponding mediation effects. No mediation effects of cg06064461 methylation showed statistical significance (one sample two-sided t-test) after correcting for multiple testing using BH-FDR method, although uncorrected significance was observed between family affirmation and peak GMV. Childexp, child's experience of family life; FamStress, family stressors; CTQ, Childhood Trauma Questionnaire. **e** Mediation model was conducted to analyse the direct and indirect effect of family affirmation on peak GMV, with cg06064461 methylation as the mediator. Results showed that cg06064461 methylation mediated the relationship between family affirmation and peak GMV with an unadjusted p value of 0.048 (one sample two-sided t-test: $\beta = 0.005$, *mediation proportion* = 9.26%, $P_{unadj} = 0.048$, $P_{adj} = 0.191$). Relevant source data were provided in the Source Data file.

mapped to *ATF2* and *MIR933* on chromosome 2. *ATF2* encodes a transcription factor of the activator protein-1 family, is ubiquitously expressed in the brain and was found to be associated with both neurodegeneration and neurogenesis[43–45]. *MIR933* shares a common promoter with *ATF2* and offers neuroprotection against neurodegenerative diseases by regulating brain-derived neurotrophic factor[46]. To validate the EWAS results, we correlated the methylation of cg06064461 with estimated GMV trajectory and peak GMV in Groups 1/2, and calculated the mediation effects of cg06064461 methylation in the adverse environment - neurodevelopment pathway. Consistent with the EWAS results, positive correlation between cg06064461 methylation and total GMV trajectory ($r = 0.14$, $P < 0.001$) (Fig. 4b) and negative correlation between cg06064461 methylation and peak GMV ($r = -0.07$, $P = 0.020$) (Fig. 4c) were observed. Overall, no mediation effects of cg06064461 methylation on the environment - neurodevelopment pathway showed statistical significance after correcting for

multiple testing (Fig. 4d and Supplementary Table 12). However, given that only one site could be identified with differential methylation between Groups 1 and 2 with relatively small sample size, it should be noted that higher level of family affirmation was associated with higher peak GMV through reduced cg06064461 methylation with an unadjusted $p$ value of 0.048 ($\beta = 0.005$, *mediation proportion* = 0.09, $P_{unadj} = 0.048$, $P_{adj} = 0.191$) (Fig. 4e). Family affirmation was defined as behaviors implemented by a parent to provide support or assistance to their children in diverse situations, demonstrating approval and affection and contributing to the parent-child relationship[47]. These results indicated that environmental exposure could potentially contribute to disadvantaged neurodevelopment and neurocognition by inducing epigenetic differences of neurogenesis-related genes. However, only a small mediation proportion was identified. Furthermore, no significant site was identified in the EWAS investigating Group 3 versus Groups 1/2 (Supplementary Fig. 14).

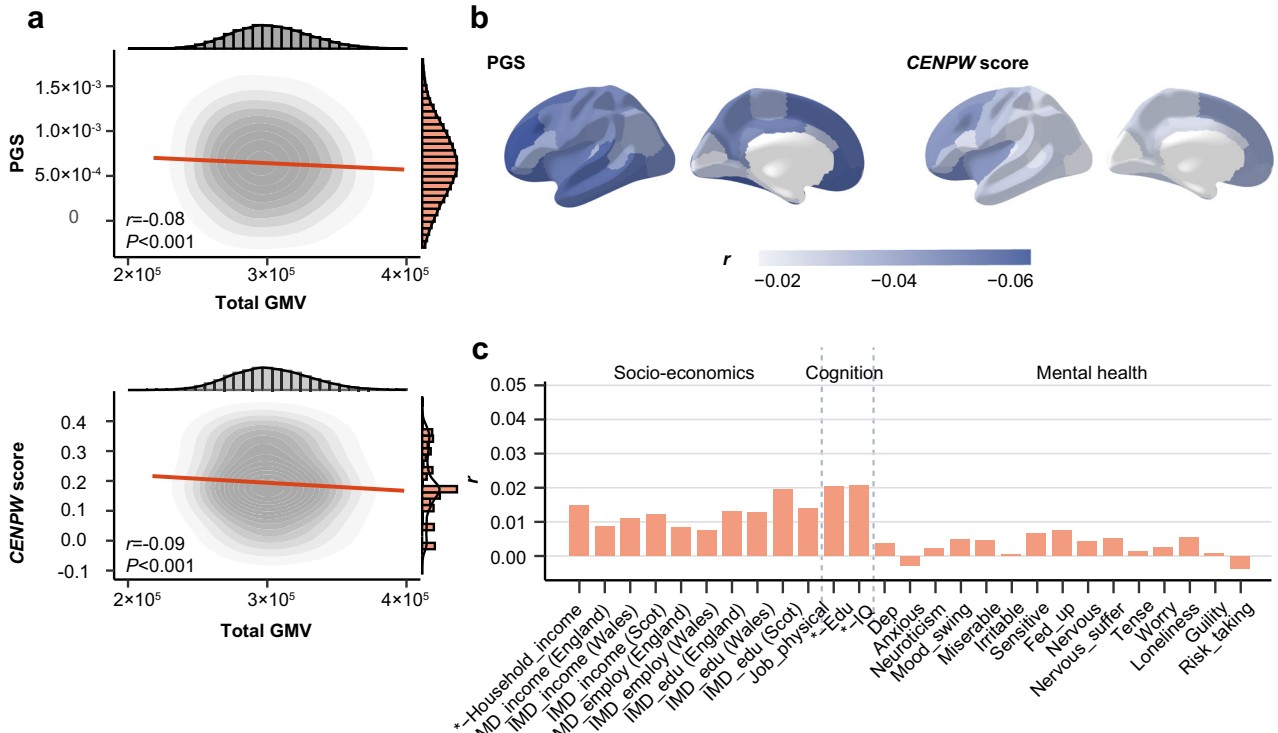

**Fig. 5 | Genetically-predicted neurodevelopment had limited impact on socio-economic, cognitive and mental health outcomes in mid-to-late adulthood.**
**a** Correlation between polygenic score (PGS) and CENPW score of delayed neurodevelopment and total gray matter volume (GMV) for participants in UK Biobank ($n = 337,199$). Marginal distributions of PGS and total GMV were both normal. PGS and CENPW score both showed negative correlation with total GMV ($r = −0.08$, $P < 2.2 \times 10^{-16}$ for PGS and $r = −0.09$, $P < 2.2 \times 10^{-16}$ for CENPW score). One sample t-test was used. PGS were averaged over different $P$ value thresholds. **b** Correlation between averaged PGS of delayed neurodevelopment and CENPW score and regional GMV for participants in UK Biobank. Rostral middle frontal ($r = −0.07$, $P_{adj} < 0.001$), fusiform ($r = −0.07$, $P_{adj} < 0.001$), lateral orbitofrontal ($r = −0.07$, $P_{adj} < 0.001$), medial orbitofrontal ($r = −0.06$, $P_{adj} < 0.001$) and rostral anterior cingulate ($r = −0.06$, $P_{adj} < 0.001$) were among the ROIs with the strongest correlation with PGS, while lateral orbitofrontal ($r = −0.06$, $P_{adj} < 0.001$), caudal middle

frontal ($r = −0.05$, $P_{adj} < 0.001$), rostral middle frontal ($r = −0.05$, $P_{adj} < 0.001$), insula ($r = −0.05$, $P_{adj} < 0.001$) and superior frontal ($r = −0.05$, $P_{adj} < 0.001$) were ROIs having the strongest correlation with CENPW score. These were consistent with the results that participants with higher PGS of delayed neurodevelopment also had worse performance in spatial working memory in UK Biobank. One sample t test was used with FDR for multiple correction. **c** Inferiority test of the correlation between averaged PGS and socio-economic, cognitive and mental health outcomes indicated that polygenic risk of delayed neurodevelopment had limited effect on the long-term socio-economic, cognitive and mental health outcomes. Full results were displayed in Supplementary Fig. 15. Similar results were observed between CENPW score and these long-term outcomes (Supplementary Fig. 16). IMD, Indices of Multiple Deprivation; Scot, Scotland; Edu, the highest educational attainment; IQ, intelligence. Relevant source data were provided in the Source Data file.

## Genetic variation had limited effects on the cognitive, mental health and socio-economic outcomes in mid-to-late adulthood

Both genetic vulnerability and structural neurodevelopment are well-established to have profound impact on one's physical, social, and mental well-being in mid-to-late adulthood[48,49]. However, the neuro-biological mechanisms through which the long-term effects of genetic variation could be manifested remain largely unknown. We conclude our study by testing in the UKB whether, and to what degree, polygenic risk for delayed neurodevelopment could have impact on the long-term brain structure, cognition, social-economic outcomes, and mental well-being. Here, socioeconomic conditions were assessed by average total household income, physical labor involvement, and Indices of Multiple Deprivation in the aspects of education, employ-ment, and income. Cognition was assessed by fluid intelligence and the highest educational attainment. Mental health was assessed by the diagnosis of anxiety and depression, neuroticism score, and self-reported mental symptoms such as the stability of emotions and the occurrence of negative emotions.

Motivated by the Group 3 GWAS results, we first calculated the PGS and CENPW score of delayed neurodevelopment for each parti-cipant in UKB and then correlated them with outcomes of interest. Both PGS and CENPW score were approximately normally distributed

and negatively associated with total GMV among this population (Fig. 5a). Next, we inspected the association of PGS and CENPW score with GMV in multiple brain regions in UKB and found that rostral middle frontal, fusiform, lateral orbitofrontal, medial orbitofrontal and rostral anterior cingulate areas were among the most correlated ROIs with PGS of delayed neurodevelopment, and lateral orbitofrontal, caudal middle frontal, rostral middle frontal, insula and superior frontal areas were among the most correlated ROIs with CENPW score (Fig. 5b and Supplementary Tables 13, 14). These were consistent with the worse spatial working memory among participants with higher PGS of delayed neurodevelopment and CENPW score (Supplementary Table 15). Findings of a negative correlation between PGS and lower GMV in these regions could be interpreted as either continued influ-ence of delayed neurodevelopment, effects from genetically-related environmental exposures or genetically-related neurodegenerative processes. Further studies are needed to explore and disentangle the potential underlying biological mechanisms. Finally, we conducted non-superiority tests of the correlation coefficients and found that correlations between PGS of delayed neurodevelopment and CENPW score and all outcomes of interest were smaller than 0.05 (Fig. 5c and Supplementary Figs. 15, 16). This indicated that although polygenic risks were related with delayed neurodevelopment during

adolescence, their long-term influences on the cognitive, mental health and socio-economic outcomes were limited once neurocognitive abilities were fully developed.

## Discussion

Adolescence is a dynamic maturational period characterized by potentially suboptimal decision making and an amplified risk of behavioral problems due to the immature brain and cognitive abilities[50–52]. There is a growing consensus that adolescents have remarkably heterogeneous brain developmental patterns[13]. Therefore, studies at the population average level may obscure the true relationship between dynamic brain changes and risks for neuropsychiatric disorders. Here, we developed a data-driven approach that identified three groups of adolescents with distinct whole-brain neurodevelopmental patterns, and showed that these groups had associated genetic or epigenetic variations, and could be related with the paths of both neurocognitive development and long-term socio-economic attainments and mental well-being in mid-to-late adulthood.

Both neuroimaging and animal studies show that gray matter in higher-order brain regions undergoes continuous thinning during adolescence with synaptic pruning and myelination[18,21]. Therefore, increasing gray matter during this period, especially in higher-order brain regions responsible for executive functions, is indicative of delayed brain maturation. Furthermore, a slower rate of gray matter thinning suggests reduced density of synapses and myelination[3], which would further limit the enhancement of neurocognitive function and efficient information processing[53–55]. These diverse growth trajectories of the adolescent brain are capable of shifting both behaviors and the learning capabilities, in ways that could lead to life-long impacts[52,56–58]. Further, human brain development involves continuing and complex interactions between genetic and environmental influences[16,19,59,60]. By integrating genomic, neuroimaging, behavior, and health-related data from three large-scale population cohorts, we confirmed that genetic variants are associated with delayed brain maturation and neurocognitive development, without affecting the socio-economic and mental well-being later in life. Whereas, adverse environmental exposure and the associated epigenetic variations were related with prolonged negative effects on brain development and behavioral disadvantages. Importantly, we regard the differences between Group 2 and Group 1 as quantitative and subject to the magnitude of cumulative adverse environmental exposure, as reflected by the large overlap in their neurodevelopmental patterns and the relatively small effect sizes associated with adverse environmental factors. Consolidating results from EWAS and mediation analysis, our study shed light on the possible epigenetic and neurobiological mechanisms underlying potential causal pathways between environmental exposure and adolescent brain development. However, it does not necessarily mean that the differences between Group 3 and Group 1 could only be attributed to genetic variation, or that differences between Group 2 and Group 1 was purely due to environment. Future research with larger sample sizes and adequate statistical power are needed to elucidate the potential interplay between gene and environment on structural brain development.

Overall, this work investigated longitudinal brain development at the individual level and its associations with neurocognition and socio-economic outcomes persisting into late adulthood. The three population cohorts involved in our analyses were designed for relatively different purposes, in different populations and produced different data components. Although we tried to link the neurodevelopmental patterns from IMAGEN to ABCD and UKB, this mapping using genetic and neuroimaging associations may subject to confounding bias. For example, the bridging between IMAGEN and ABCD assumed a linear change of GMV 9 years old (baseline age for the majority participants in ABCD) to 14 years old (baseline age for the majority participants in IMAGEN), and homogenous population composition

between these two cohorts. Given the findings from existing investigations[21], a linear trend of GMV from 9 to 14 years old were attainable, and in order to achieve population homogeneity, we only selected participants of self-reported "white" ethnicity in ABCD. In addition, both the appropriateness of using the proxy phenotype and results of GWAS conducted in ABCD were successfully validated. However, the robustness of the bridge approach used in this study and its assumptions still await further validation once follow-up data become available for the ABCD participants. Meanwhile, long-term follow-ups of the socio-economic outcomes in IMAGEN adolescents are needed to validate our results obtained from UK Biobank. In other words, large-scale longitudinal data that span the entire life-course may confirm the reliability of the findings obtained in our study. Further, the IMAGEN study involves healthy individuals only and our findings may have limited generalizability to specific disease populations. Although these adolescents were not diagnosed with specific neuropsychiatric disorders at baseline, they were likely to be present with subclinical symptoms, referred to minor neurological abnormalities or dysfunction seen in the absence of an obvious cause or pathology. Evidence indicated that subclinical symptoms seen in normal young children were partly attributable to immaturity of the nervous system and were frequently found in the clinical course of psychosis[61], schizophrenia[62] and Alzheimer's disease[63]. Neuroimaging studies thus stand as a powerful tool for identifying important brain regions and morphological phenotypes associated with subclinical symptoms, and for elucidating the neurobiological correlates of subclinical symptoms along the course of brain development. Finally, the three groups identified in this study constitute an initial attempt to solve the problem of heterogeneous brain development that relies heavily on the image-derived phenotypes obtained from sMRI. Further investigation using other neuroimaging modalities, or multi-modal phenotypes are needed for a comprehensive understanding of this dynamic process.

## Methods
### Ethical statement

All the cohort data used in this study complies with relevant ethical regulations. ABCD and Human Connectome Project (HCP) study was supported by the National Institutes of Health (NIH). The Philadelphia Neurodevelopmental Cohort (PNC) was approved by both the University of Pennsylvania and the Children's Hospital of Philadelphia. The IMAGEN study was approved by local ethical research committees at each research site: King's College London, University of Nottingham, Trinity College Dublin, University of Heidelberg, Technische Universitat Dresden, Commissariat a l'Energie Atomique et aux Energies Alternatives, and University Medical Center. UK Biobank has approval from the North West Multi-centre Research Ethics Committee as a Research Tissue Bank approval. Informed consent was sought from all participants and a parent/guardian of each participant if under 18 years in all studies.

### Participants

Genomic, neuroimaging, environmental exposure, behavioral and mental health related data used to identify adolescent neurodevelopmental patterns were obtained from the IMAGEN study. Individuals with GMV beyond 4 interquartile ranges (IQRs) in any ROI were considered as outliers and were excluded from the analyses. After applying the exclusion criteria, 1543 adolescents (48.4% males) with at least two structural MRI scans from 14 to 23 years old were included in the analyses (Supplementary Table 16). The average number of structural MRI scans per participant was 2.63, with 974 adolescents having a total of 3 scans (at 14, 19, and 23 years, respectively) and 569 adolescents having a total of 2 scans (384 at 14 y and 19 y, 147 at 14 y and 23 y and 38 at 19 y and 23 y). In addition, genotyping data used in GWAS, validation of GWAS and investigation of the long-term impact were obtained

from ABCD, IMAGEN and UKB, respectively. A total of 11,760 participants (52.2% males) at baseline aged between 9 and 11 years old from ABCD were included, with the average number of structural MRI scans per adolescent 1.68. Further, 502,409 participants aged between 37 and 73 years old with 45.6% males from UKB were included in the long-term analyses of structural brain development. Demographics and baseline characteristics of participants from the three large population cohorts were summarized in Supplementary Table 17. Additionally, a total of 652 participants aged 5–22 (46.2% males) in HCP Development (HCP-D), a total of 1113 participants aged 22–37 (45.6% males) in HCP Young Adult (HCP-YA) and a total of 1587 participants aged 8–23 (47.6% males) in PNC were included only for the neuroimaging analysis. A full description of all population cohorts used in the analyses can be found in Supplementary Methods.

### Analysis of structural MRI data

**Data preprocessing.** In brief, quality-controlled processed T1-weighted neuroimaging data were obtained from ABCD, IMAGEN, HCP-D, HCP-YA, and PNC. Assessment of regional morphometric structure were extracted by FreeSurfer v6.0 cross-sectional pipelines using Desikan-Killiany (h.aparc) atlas for cortical regions, and ASEG atlas for subcortical regions. Quality check was performed according to FreeSurfer reconstruction quality-controlled (QC) measures. Detailed description of data collection and preprocessing is provided in Supplementary Methods.

**Estimation of GMV developmental trajectory.** GMV trajectory in each of the 44 ROIs was estimated for each adolescent using linear mixed effect regression model (*lme4* 1.1-31 package) (since at most three structural MRI scans were available for each adolescent, only random slope model could be robustly estimated). Empirical Bayes estimate of the random slope was extracted for each adolescent. Intracranial volume (ICV), sex, handedness and imaging site were used as covariates to adjust for potential confounding.

**Principal component analysis (PCA) and group clustering.** Dimension reduction via PCA (*prcomp* function in the *stats* 4.2.2 package) was performed on standardized individual GMV trajectories estimated using neuroimaging data of 44 ROIs. The rotation matrix was obtained from the right singular vector, where singular value decomposition was performed on the centered GMV trajectories. Considering both the proportion of cumulative variance explained and robustness of the multivariate clustering results, the first 15 PCs (Supplementary Table 1), which explained 80% of the total variation, were used in the multivariate k-means clustering. The optimal number of clusters was selected based on the Elbow method with the constraint that each cluster contain at least 4% of the overall population.

**Permutation test.** Permutation was conducted by shuffling the estimated GMV trajectory in each ROI simultaneously and re-performing the dimension reduction and multivariate clustering repeatedly over 1000 times. *P* value was calculated as the proportion of Between-cluster Sum of Squares/Total Sum of Squares ratio greater than the estimated ratio in the original sample across all 1000 permutations.

**Comparison of GMV trajectory among groups.** Pairwise comparisons of GMV developmental trajectories in each ROI among the three groups were conducted via t test. The top 5 ROIs with the largest absolute t values were selected as the top distinguishing ROIs between the corresponding groups (Supplementary Table 3). Cohen's ds (calculated using *effectsize* 0.8.3 package) for these regions were provided in Fig. 1d and Supplementary Fig. 17.

**Estimation of age and region-specific GMV development among groups.** To illustrate the region-specific GMV development in

an extended time frame ranging from late childhood to early adulthood, external neuroimaging data from several population cohorts were incorporated. This includes a total of 21,826 participants comprising of 11,811 participants aged 9–14 y with 19,587 scans in ABCD, 652 participants aged 5–22 y in HCP-D, and 1587 participants aged 8–23 y in PNC study. Since cubic model could not capture GMV trajectory beyond 23 y and quadratic model could not utilize data before 9 y, we used a reference curve estimated from cross sectional studies (HCP-D + PNC) in estimating the region-specific GMV developmental curve over 5–25 y. Distance between GMV in IMAGEN and that in the reference population in the corresponding ROI was used as the dependent variable in the quadratic linear mixed effect model with random intercept and slope, adjusting for ICV and site. Empirical Bayes estimates of the random effects for each group were added to the population averaged estimates to yield the group-specific developmental curve.

**Estimation of group-specific developmental curve of total GMV in IMAGEN.** A two-stage estimating procedure was adopted. Optimal model was selected among a series of polynomial mixed effect models using likelihood ratio test. First, population ICV developmental curve over 5–23 y was estimated using the above-mentioned population neuroimaging data. Quadratic linear mixed effect models with random intercept at the individual and study level were fitted. To estimate the developmental curve of total GMV in the reference population (ABCD + HCP + PNC), cubic model adjusting for ICV was selected with random intercepts at the individual and study level. To estimate the developmental curve of total GMV in the ABCD and IMAGEN population, cubic model adjusting for ICV was selected with random intercept and slope at the individual level. Empirical Bayes estimates of the random effects were extracted and averaged in each group. Population ICV estimated in stage 1 was used to fit group-specific curves. The 5th and 95th percentile of the group-specific total GMV were calculated as the 95% confidence interval at each age.

**Estimation of peak total GMV in IMAGEN.** To estimate the peak total GMV in the IMAGEN population, 1113 participants aged 22–38 y in HCP-YA study were added to the reference population. A similar two-stage estimating procedure was used and the optimal model was selected based on Bayesian information criterion (BIC) and likelihood ratio test. First, population developmental curve of ICV was estimated using mixed effect regression model (*nlme* 3.1-160 package) with random intercept and slope. Basis function involving centered age was determined as the natural spline with 5 degrees of freedom. Interaction effects between age and sex, sex, and study were also included in the regression model. Estimated ICV at each age was retained for the following analysis. Next, linear mixed effect regression model with random intercept and slope was fitted for total GMV. Basis function involving centered age was determined to be B spline with 12 degrees of freedom. Interaction effects between age and sex, sex and study were included in the regression model. Peak total GMV was defined as the highest total GMV one can achieve during brain maturation.

**Comparisons of environmental burden, neurocognition, behavior and mental disorder.** To assess whether environmental burden, neurocognition, behavioral risk factors and mental symptoms differ by groups, we analyzed their longitudinal measurements at 14 y, 16 y, 19 y and 23 y in IMAGEN. Personal traits, including personality, temperament and characters, were obtained from the NEO Five-Factor Inventory (NEO-FFI) and temperament and character inventory (TCI-R). Environmental burden, including prenatal exposures (parental smoking, maternal drinking, and maternal medical problems during pregnancy), birthweight, stressful life events, child trauma experiences, child's experience of family life and family stressors, were obtained from Pregnancy and Birth Questionnaire (PBQ), life-events

questionnaire (LEQ), Childhood Trauma Questionnaire (CTQ), and Family Stress Scale and Family Life Questionnaire from development well-being assessment interview (DAWBA). Neurocognitive performances were obtained from Cambridge Neuropsychological Test Automated Battery (CANTAB) tests, Monetary-Choice Questionnaire (KIRBY) and Stop Signal Task (SST) results. Behavioral assessments, including conduct problems and substance use, were obtained from strengths and difficulties questionnaire (SDQ), European school survey project on alcohol and drugs (ESPAD), Fagerstrom test for nicotine dependence (FTND). Mental health conditions, including ADHD and depression, were obtained from self-rated development well-being assessment interview (DAWBA), where ADHD score was additionally calculated using parent-rated interview. A detailed description of these assessment instruments is provided in Supplementary Methods. Generalized linear models adjusting for sex, handedness and ICV were used for comparing these tests at baseline and at the last follow-up visit (19 y for Pattern recognition memory, Affective Go-No Go (AGN) and Rapid visual information processing (RVP); 23 y for all other tests). For child-rated ADHD and depression score, baseline scores were also included as covariates. Intra-Extra Dimensional Set Shift (IED) test was only available at age 23 y, and parent-rated ADHD score was only available at 14 y and 16 y. Cohen's d was calculated for each measurement after regressing out the covariates. False discovery rate (FDR) was used to correct for multiple testing within scales.

**Quality control of genomic data.** In this study, we performed stringent QC standards using PLINK 1.90. Individuals with >10% missing rate and SNPs with call rates <95%, minor allele frequency <0.1%, deviation from the Hardy-Weinberg equilibrium with $P < 1E\text{-}10$ were excluded from the analysis. For ABCD, we only selected subjects with self-reporting white ancestral origins using the public release 3.0 imputed genotype data, which was imputed with the HRC reference panel[64]. Considering that ABCD is oversampled for siblings and twins, we randomly selected one participant within each family. For IMAGEN, details about preprocessing of genomic data can be found in previous reports[65] and data was imputed with the HapMap3 reference panel[66]. For UKB, we selected subjects that were estimated to have recent British ancestry and have no more than ten putative third-degree relatives in the kinship table using the sample quality control information provided by UKB. For more details, please refer to the official document for genetic data of the UKB. After quality control, we obtained a total of 5,020,358 SNPs and 7662 participants in ABCD, 5,966,316 SNPs and 1982 participants in IMAGEN, and 616,339 SNPs and 337,199 participants in UKB.

**Calculation of genetic liability.** For each individual, PGS of ADHD, ASD, EA, and IQ were calculated based on the public GWAS summary statistics[39,67–69] using PRSice v2.3.3. For ADHD, ASD, and IQ, optimal p value thresholds were determined based on the best-fit $R^2$ using parent-rated psychiatric scores for ADHD and ASD, and the total WISCIV score (Supplementary Fig. 18). For EA, variants were selected using a P value threshold from 5e-08 to 1 with a step of 5e-05 and an average score under each P value threshold was calculated. One-way ANOVA test with Fisher's Least Significant Difference (LSD) post hoc test was used to compare PGS among groups.

**GWAS and validation.** Since it was difficult to estimate individual GMV developmental trajectory in ABCD with limited number of structural MRI scans per participant and limited age range, we calculated the group-reweighted GMV as a proxy phenotype. There are several underlying assumptions in this calculation. Firstly, it assumes that all brain regions exhibit a comparable linear change from childhood to adolescence. Secondly, it assumes that the participants from ABCD and IMAGEN are drawn from a homogeneous population. Once again, we only included individuals in ABCD with self-report white ancestral

origins. ROI-specific loading contributing to group classification (Group 2 vs Group 1, Group 3 vs Group 1, Group 3 vs Group 2, and Group 3 vs Groups 1/2) were obtained by regressing baseline GMV in 44 ROIs adjusting for age, sex, handedness, and site in IMAGEN. Logistic regression model was used as the classification model and top 10 ROIs with the largest loadings were used to calculate the group-reweighted GMV in ABCD using only baseline data. Since results remained similar when comparing Group 3 vs Group 1 and when comparing Group 3 vs Group 2 (Supplementary Figs. 19, 20), we combined Group 1 and 2 for increased statistical power, and performed the GWAS to investigate the genetic variations associated with Group 3 vs Groups 1/2 (delayed brain development). GWAS was conducted in the white population adjusting for sex, scanner effect and top 20 PCA components using Plink 2 (Supplementary Fig. 21). To ensure the validity of group reweighted phenotype, we correlated the Group-3 reweighted GMV and neurocognitive assessments consisting of Game of Dice Task (GDT), Delay Discounting Task (DDT) and NIH Toolbox in ABCD. The NIH Toolbox includes Picture Vocabulary Test (PVT), Flanker Inhibitory Control and Attention Test (Flanker), List Sorting Working Memory Test (List), Dimensional Change Card Sort Test (DCCS), Pattern Comparison Processing Speed Test (Pattern), Picture Sequence Memory Test (PSMT), Oral Reading Recognition Test (Reading) and provided fluid cognition composite score (FluidCog), crystallized cognition composite score (CrystalCog). Details for all the measurements could be found in Supplementary Methods. Gene-based association analysis was conducted via MAGMA (version 1.10) using raw genomics data with the same covariate adjustment. To validate the GWAS results, PGSs for SNPs residing in *CENPW* (referred to as *CENPW* score) and across the whole genome (referred to as PGS) were calculated. Four SNPs were obtained by clumping within 250 kb upstream and downstream of *CENPW* (chr6:126339789-126483320) using Plink 2. PGS was calculated using PRSice using the most predictive P threshold for group-reweighted baseline GMV (Supplementary Fig. 22). Distribution of PGS between Group 3 vs Groups 1/2 and correlation coefficients between PGS and neurocognition, behavior and mental disorder at age 14 y and 23 y were obtained. FDR was used for multiple tests correction within scales.

**EWAS, gene-specific methylation analysis and results validation.** EWAS was performed among Group 1 ($n = 446$), Group 2 ($n = 463$) and Group 3 ($n = 36$) in IMAGEN. Methylation data were collected using the Illumina Infinium HumanMethylation450 BeadChip. Locus-specific genome-wise methylation analysis was conducted and beta values at each Autosomal CpG site were used in pairwise comparisons with group label as the phenotype using logistic regression adjusting for sex, experimental batches (recruitment center and acquisition wave), the first two PCs of methylation composition and the first four PCs of estimated differential cell counts. We used Synthetic Minority Oversampling Technique (SMOTE) (*smote* function in *performanceEstimation* 1.1.0 package; default setting) to address the issue of class imbalance when comparing Group 3 with others. Statistical significance was set as FDR adjusted $p < 0.05$. Next, we aimed to investigate the association between CpG site and gene methylation with environmental factors of interest. We conducted mediation analyses (*sem* function in the *lavaan* 0.6-12 package) and estimated the total effect of childhood environmental exposures on estimated peak GMV and the indirect effect mediated by cg06064461 hypermethylation. Sex, batches effects, methylation composition components and differential cell count components were included as covariates. Total, direct, and indirect effects and their standard deviations were estimated using 1000-iterated nonparametric bootstrap approach. FDR was used to correct for multiple testing within scales. Childhood environmental exposures included abuse (physical/emotional/sexual) and neglect (physical/emotional) scores in the CTQ, socioeconomics/housing, work/pressure, health and relationship/addiction scores in

DAWBA-Family Stress Scale, and affirmation, discipline, rules and special allowance scores in the DAWBA-Family Life Questionnaire. The calculation details are presented in the Supplementary Methods.

**Long-term impacts of neurodevelopment in UK Biobank (UKB).** Socio-economic, cognitive, and mental health outcomes were obtained at baseline visit among participants in UKB. Socioeconomic conditions were assessed by average total household income (Field ID: 738) discretized by 18 k, 40 k, 52 k, and 100 k, jobs involved in physical activity (Field ID: 816) and Indices of Multiple Deprivation (IMD) (education, employment and income scores) (Category ID: 76). IMD scores were offered separately in England, Scotland and Wales by the UK government. Cognition was assessed by fluid intelligence (Field ID: 20016) and the highest educational attainment for individuals (Field ID: 6138). The educational attainment was divided into four ordinal categories: (1) College or University degree; (2) A levels/ AS levels, NVQ or HND or HNC, other professional qualifications or equivalent; (3) O levels/GCSEs, CSEs or equivalent; (4) None of the above. Mental health was assessed by diagnosis of anxiety and depression (Field ID: 41270), neuroticism score (Field ID: 20127) and self-reported mental symptom appearances, including mood swing (Field ID: 1920), miserableness (Field ID: 1930), irritability (Field ID: 1940), sensitivity (Field ID: 1950), fed-up feelings (Field ID: 1960), nervous feelings (Field ID: 1970), suffer from 'nerves' (Field ID: 2010), tense feelings (Field ID: 1990), worrier feelings (Field ID: 1980), loneliness (Field ID: 2020), guilty feelings (Field ID: 2030) and tendency to take risks (Field ID: 2040). A detailed description of assessment instruments used in the analysis can be found in Supplementary Methods. To estimate the long-term effect of delayed neurodevelopment, we calculated *CENPW* score and RPS according to the results of Group 3 GWAS and correlated these scores with outcomes of interest after regressing out the age effect at recruitment, site and gender. It should be noted that these scores only reflect a genetic predicted risk for delayed brain development. Given the large age gap between participants in UKB and IMAGEN, it is challenging to disentangle the long-term impacts of neurodevelopment from those due to potential environmental confounding in mid-to-late adulthood. Therefore, this analysis only serves to explore the potential long-term influence of genetically predicted delayed neurodevelopment and does not account for potential confounding due to environmental factors. Similarly, we assume the homogeneity of study populations between IMAGEN and UKB. For PGS calculation, we used $P$ value thresholds from 5E-08 to 1 with a step of 5E-05 and calculated an average PGS score for each individual. Due to the large sample size and easily-obtainable statistical significance, inferiority tests against 0.05 were conducted against the null hypothesis that the absolute correlation coefficient was less than 0.05.

### Reporting summary
Further information on research design is available in the Nature Portfolio Reporting Summary linked to this article.

## Data availability
The summary statistics of the GWAS for delayed brain development generated in this study has been deposited in the NHGRI-EBI Catalog of human GWAS database (https://www.ebi.ac.uk/gwas/) under GCP ID GCP000904 upon publication or is available at https://delayedneurodevelopment.page.link/amTC. The raw ABCD, IMAGEN, HCP, PNC and UKB data are protected and are not available due to data privacy laws. However, access can be obtained upon application. ABCD data can be accessed at https://abcdstudy.org/; IMAGEN data can be accessed by email at https://imagen-project.org/; HCP data are available from: https://www.humanconnectome.org/; PNC data can be accessed from dbGaP: https://www.ncbi.nlm.nih.gov/projects/gap/cgi-bin/study.cgi?study_id=phs000607.v3.p2; and UKB data can be accessed at https://biobank.ndph.ox.ac.uk/. Public GWAS summary statistics of ADHD and ASD used in this study are available in the Psychiatric Genomics Consortium database of summary statistics at https://www.med.unc.edu/pgc/results-and-downloads, while public GWAS summary statistics of EA can be accessed at http://www.thessgac.org/data and public GWAS summary statistics of IQ can be accessed at https://ctg.cncr.nl/. All the data generated in this study are provided in the Supplementary Information and Source Data file. Source data are provided with this paper.

## Code availability
Primary analyses were conducted in R v4.2.2. Linear mixed effect models were performed using *lme4* 1.1-31 and *nlme* 3.1-160R packages. Mediation analysis was performed using *lavaan* 0.6-12R package. PLINK 2.0 was used to perform GWAS and calculate *CENPW* score. MAGMA v1.10 was used to perform the gene-based association analysis. PRSice v2.3.3 was used to calculate the PGS. Custom code that supports the main findings of this study was available at https://github.com/abnmsry/Life-course-investigation-of-structural-neurodevelopment-at-the-individual-level[70]. Additional information related to this paper are available from the authors upon reasonable request.

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

## Acknowledgements

This work received support from the following sources: the National Nature Science Foundation of China (No.82304241 [to X.L.]), the General Projects of Shanghai Science and Technology Commission (No. 21ZR1405000 [to X.L.]), National Key R&D Program of China (No.2019YFA0709502 [to J.F.], No.2018YFC1312904 [to J.F.]), Shanghai Municipal Science and Technology Major Project (No.2018SHZDZX01 [to J.F.], ZJ Lab [to J.F.], and Shanghai Center for Brain Science and Brain-Inspired Technology [to J.F.]), the 111 Project (No.B18015 [to J.F.]), the European Union-funded FP6 Integrated Project IMAGEN (Reinforcement-related behavior in normal brain function and psychopathology) (LSHM-CT- 2007-037286 [to G.S.]), the Horizon 2020 funded ERC Advanced Grant 'STRATIFY' (Brain network based stratification of reinforcement-related disorders) (695313 [to G.S.]), Human Brain Project (HBP SGA 2, 785907, and HBP SGA 3, 945539 [to G.S.]), the Medical Research Council Grant 'c-VEDA' (Consortium on Vulnerability to Externalizing Disorders and Addictions) (MR/N000390/1 [to G.S.]), the National Institute of Health (NIH) (R01DA049238 [to G.S.], A decentralized macro and micro gene-by-environment interaction analysis of substance use behavior and its brain biomarkers), the National Institute for Health Research (NIHR) Biomedical Research Centre at South London and Maudsley NHS Foundation Trust and King's College London, the Bundesministeriumfür Bildung und Forschung (BMBF grants 01GS08152; 01EV0711 [to G.S.]; Forschungsnetz AERIAL 01EE1406A, 01EE1406B; Forschungsnetz IMAC-Mind 01GL1745B [to G.S.]), the Deutsche Forschungsgemeinschaft (DFG grants SM 80/7-2, SFB 940, TRR 265, NE 1383/14-1 [to G.S.]), the Medical Research Foundation and Medical Research Council (grants MR/R00465X/1 and MR/S020306/1 [to S.D.]), the National Institutes of Health (NIH) funded ENIGMA (grants 5U54EB020403-05 and 1R56AG058854-01 [to S.D.]), NSFC grant 82150710554 and environMENTAL grant. Further support was provided by grants from: - the ANR (ANR-12-SAMA-0004, AAPG2019 - GeBra [to J.-L.M.]), the Eranet Neuron (AF12-NEUR0008-01 - WM2NA; and ANR-18-NEUR00002-01 - ADORe [to J.-L.M.]), the Fondation de France (00081242 [to J.-L.M.]), the Fondation pour la Recherche Médicale (DPA20140629802 [to J.-L.M.]), the Mission Interministérielle de Lutte-contre-les-Drogues-et-les-Conduites-Addictives (MILDECA [to J.-L.M.]), the Assistance-Publique-Hôpitaux-de-Paris and INSERM (interface grant [to M.-L.P.M.]), Paris Sud University IDEX 2012 [to J.-L.M.], the Fondation de l'Avenir (grant AP-RM-17-013 [to M.-L.P.M.]), the Fédération pour la Recherche sur le Cerveau; the National Institutes of Health, Science Foundation Ireland (16/ERCD/3797 [to R.W.]) and U.S.A. (Axon, Testosterone and Mental Health during Adolescence; RO1 MH085772-01A1 [to T.P.]) and by NIH Consortium grant U54 EB020403 [to S.D.], supported by a cross-NIH alliance that funds Big Data to Knowledge Centres of Excellence. The funders had no role in study design, data collection and analysis, decision to publish or preparation of the manuscript.

## Author contributions

X.L., T.J., J.F., and B.J.S. conceptualized the study; R.S., X.L., and T.J. designed the analytic approach; R.S. analysed the data and visualized the results; R.S. and X.L. wrote the manuscript; S.X. and J.K. helped in preprocessing the neuroimaging and genetic data; S.X., T.J. B.J.S., and J.F. helped in interpreting the results; T.W.R., T.B., G.J.B., S.D., G.S., B.J.S., and J.F. revised the first draft; T.B., G.J.B., A.L.W.B., S.D., H.F., A.G., H.G., P.G., A.H., R.B., J.-L.M., M.-L.P.M., E.A., F.N., D.P.O., T.P. L.P., S.H, S.M., J.H.F., M.N.S., N.V., H.W., R.W., and G.S. were the principal investigators of IMAGEN Consortium; Imaging, genetic and behavioral data in the IMAGEN project were acquired and provided by the IMAGEN Consortium; All authors critically revised the manuscript. X.L., B.J.S., and J.F. contributed equally to this paper.

## Competing interests

The other authors declare no competing interests. Dr Banaschewski served in an advisory or consultancy role for Lundbeck, Medice, Neurim Pharmaceuticals, Oberberg GmbH, Shire. He received conference support or speaker's fee by Lilly, Medice, Novartis and Shire. He has been involved in clinical trials conducted by Shire & Viforpharma. He received royalties from Hogrefe, Kohlhammer, CIP Medien, Oxford University Press. The present work is unrelated to the above grants and relationships. Dr Barker receives honoraria for teaching from GE Healthcare. Dr Poustka served in an advisory or consultancy role for Roche and Viforpharm and received speaker's fee by Shire. She received royalties from Hogrefe, Kohlhammer and Schattauer. The present work is unrelated to the above grants and relationships. The other authors declare no competing interests.

## Additional information

[1]School of Data Science, Fudan University, Shanghai, China. [2]Institute of Science and Technology for Brain-Inspired Intelligence, Fudan University, Shanghai, China. [3]Key Laboratory of Computational Neuroscience and Brain-Inspired Intelligence (Fudan University), Ministry of Education, Shanghai, China. [4]Social Genetic and Developmental Psychiatry Centre, Institute of Psychiatry, Psychology and Neuroscience, King's College London, London, UK. [5]Centre for Population Neuroscience and Precision Medicine (PONS), Institute of Science and Technology for Brain-Inspired Intelligence (ISTBI), Fudan University, Shanghai, China. [6]School of Psychology, University of Southampton, Southampton, UK. [7]Department of Psychology and Behavioural and Clinical Neuroscience Institute, University of Cambridge, Cambridge, UK. [8]Department of Child and Adolescent Psychiatry and Psychotherapy, Central Institute of Mental Health, Medical Faculty Mannheim, Heidelberg University, Square J5, Mannheim, Germany. [9]Department of Neuroimaging, Institute of Psychiatry, Psychology & Neuroscience, King's College London, London, UK. [10]Discipline of Psychiatry, School of Medicine and Trinity College Institute of Neuroscience, Trinity College Dublin, Dublin, Ireland. [11]Institute of Cognitive and Clinical Neuroscience, Central Institute of Mental Health, Medical Faculty Mannheim, Heidelberg University, Square J5, Mannheim, Germany. [12]Department of Psychology, School of Social Sciences, University of Mannheim, Mannheim, Germany. [13]NeuroSpin, CEA, Université Paris-Saclay, F-91191 Gif-sur-Yvette, France. [14]Departments of Psychiatry and Psychology, University of Vermont, Burlington, VT, USA. [15]Sir Peter Mansfield Imaging Centre School of Physics and Astronomy, University of Nottingham, University Park, Nottingham, UK. [16]Department of Psychiatry and Psychotherapy CCM, Charité-Universitätsmedizin Berlin, corporate member of Freie Universität Berlin, Humboldt-Universität zu Berlin, and Berlin Institute of Health, Berlin, Germany. [17]Physikalisch-Technische Bundesanstalt (PTB), Braunschweig and Berlin, Germany. [18]Institut National de la Santé et de la Recherche Médicale, INSERM U A10 "Trajectoires développementales en psychiatrie", Université Paris-Saclay, Ecole Normale supérieure Paris-Saclay, CNRS, Centre Borelli, Gif-sur-Yvette, France. [19]Department of Child and Adolescent Psychiatry, AP-HP, Sorbonne Université, Pitié-Salpêtrière Hospital, Paris, France. [20]Psychiatry Department, EPS Barthélémy Durand, Etampes, France. [21]Institute of Medical Psychology and Medical Sociology, University Medical Center Schleswig-Holstein Kiel University, Kiel, Germany. [22]Department of Psychiatry, Faculty of Medicine and Centre Hospitalier Universitaire Sainte-Justine, University of Montreal, Montreal, QC, Canada. [23]Departments of Psychiatry and Psychology, University of Toronto, Toronto, ON, Canada. [24]Department of Child and Adolescent Psychiatry and Psychotherapy, University Medical Centre Göttingen, von-Siebold-Str. 5, Göttingen, Germany. [25]Department of Psychiatry and Neuroimaging Center, Technische Universität Dresden, Dresden, Germany. [26]Centre for Population Neuroscience and Stratified Medicine (PONS), Department of Psychiatry and Psychotherapy, Charité Universitätsmedizin Berlin, Berlin, Germany. [27]School of Psychology and Global Brain Health Institute, Trinity College Dublin, Dublin, Ireland. [28]Huashan Institute of Medicine, Huashan Hospital affiliated to Fudan University, Shanghai, China. [29]MOE Frontiers Center for Brain Science, Fudan University, Shanghai, China. [30]Zhangjiang Fudan International Innovation Center, Shanghai, China. [31]Department of Computer Science, University of Warwick, Coventry, UK. [32]These authors contributed equally: Runye Shi, Shitong Xiang, Tianye Jia.
✉e-mail: xiaoleilin@fudan.edu.cn; bjs1001@cam.ac.uk; jianfeng64@gmail.com

## IMAGEN Consortium

**Tianye Jia**[2,3,4,5], **Tobias Banaschewski**[8], **Gareth J. Barker**[9], **Arun L. W. Bokde**[10], **Sylvane Desrivières**[4], **Herta Flor**[11,12], **Antoine Grigis**[13], **Hugh Garavan**[14], **Penny Gowland**[15], **Andreas Heinz**[16], **Rüdiger Brühl**[17], **Jean-Luc Martinot**[18], **Marie-Laure Paillère Martinot**[18,19], **Eric Artiges**[18,20], **Frauke Nees**[8,11,21], **Dimitri Papadopoulos Orfanos**[13], **Tomáš Paus**[22,23], **Luise Poustka**[24], **Sarah Hohmann**[8], **Sabina Millenet**[8], **Juliane H. Fröhner**[25], **Michael N. Smolka**[25], **Nilakshi Vaidya**[26], **Henrik Walter**[16], **Robert Whelan**[27] & **Gunter Schumann**[5,26]

