## [Peer Review File · Nature Communications]

Investigating grey matter volumetric trajectories through the
lifespan at the individual levelREVIEWERS' COMMENTS:

Reviewer #1 (Remarks to the Author):

Shi et al analysed sMRI data from the IMAGEN cohort and found evidence that adolescents can be clustered into three groups. This finding supports and extends previous findings of marked heterogeneity in structural neurodevelopment (e.g., ref 17-19). In addition, they used data from the ABCD, IMAGEN and UKB to study factors that could contribute to the heterogeneity, and also to predict long-term impacts in adulthood.

Overall, the team has produced an impressive set of results, and I fully support their emphasis on individual heterogeneity. Their efforts to combine and integrate various databases were creative, but as detailed below, it is also potentially a major weakness of the report.

The Introduction is well written, but presumably due to space limitations several crucial aspects of the report are not introduced. These include the use of the Cambridge Gambling Task to substantiate a differentiation between PC1 and PC2, and later on the use of ABCD and UKB cohorts in analyses of (epi)genetics and long-term outcomes.

The authors stress the use of a data-driven approach, but the analyses of developmental trajectories were based on 34 cortical and 10 subcortical ROIs. Was it for computational reasons that the analysis was limited to certain ROIs? This could be clarified and may warrant a comment that the analysis was not purely data-driven but rested on select ROIs. This issue is underscored by the use of PCA to identify a subset of ROIs. Again, it was unclear why this dimension reduction was necessary given the data-driven emphasis. Finally, in subsequent analyses (line 185 ->), the authors again refer to certain ROIs, which was quite confusing in view of the initial analytic reduction steps.

The first 15 PCs explained about 80% of the variance. Is it correct that the authors then restricted their clustering to PC1 and PC2? If so, how much variance did these two PCs account for? Is it possible that additional PCs, perhaps uncorrelated with baseline total GMV, would yield different results?

The identification of distinct clusters was based on k-means clustering and the "elbow method". Cluster 1 (N=711) and 2 (N=765) were large, whereas cluster 3 only included 67 individuals (4.3%). The authors describe the difference between the first two clusters as quantitative in nature, with lower baseline total GMV and slower rate of GMV decrease in cluster 2. Thus, in a categorical sense, cluster 3 seems to be the deviating group, and it had higher PRS for ADHD than groups 1 and 2.

However, as mentioned, group 3 was a very small group and the analyses of genetic variants and epigenetic changes were therefore conducted on the ABCD cohort. Specifically, using "group-reweighted" GMV as proxy phenotype, due to the fact that the ABCD has a limited age-range, the authors assessed "one's tendency of being in Group 3 relative to Groups 1 and 2" (lines 239-241). This step was quite unclear to me. If I got it correctly, it will be based on baseline GMV regression weights from the relevant contrasts (2-1, 3-1, 3-2) and the top 10 ROIs with the largest loadings formed the basis for the group-reweighted GMV in ABCS (aged 9 years). While this was a creative way of bridging study cohorts, I found it to rest on many assumptions which undermines the strength of conclusions from the genetic and epigenetic analyses. The same critique holds for the extension to UKB.

Taken together, the authors mention in their discussion that the attempt to link the neurodevelopmental patterns from IMAGEN to ABCS and UKB "may be subject to confounding bias". I fully agree and to me the paper might have come across as stronger had it involved the IMAGEN findings only.

A final comment. The authors conclude that the adolescents in group 3 are in critical need of support and counselling for their brain and neurocognitive abilities to better develop. Given that some differences diminished over time, I found this statement unclear.

Reviewer #2 (Remarks to the Author):

In this study the authors have clustered youth into three groups based on their developmental pattern of Grey Matter Volume. They looked for associations between these clusters, neurocognition and mental health symptoms. Next, they examined the genetic and

epigenetic associations with each cluster.

All-in-all, Lin and colleagues present an interesting and impressive manuscript. I have several comments that I hope will help improve the manuscript. My first list of comments are mainly aspects of the manuscript, and the reasoning behind it, that I believe needs to be made clearer so that it will be easier for the reader to follow. I am not familiar with EWAS, so I have not made comments on that part of the analysis.

1. I am having difficulties with understanding how the group-reweighted GMV can be derived in the ABCD given that the participants here are outside of the age-range in the IMAGEN dataset. From my understanding this was solved by adding additional cohorts to the IMAGEN sample, and then predicting developmental trajectories of gray matter volume. But these cohorts that were added were then cross-sectional? So the “developmental trajectories” estimation which is used later as a proxy is then based on cross-sectional data and not longitudinal data?

I also see that here there are 19 587 participants included from ABCD-study in making the age-specific GMV curves. To my knowledge there are not that many participants in the study, so have the authors then included multiple scans from the same participants? If this was the case, did you in any way control for the same person being included several times?

2. In the result section the strengths of the associations are not given in the text but referred to in the supplement. Some of the associations that are reported have a very low effect size even though they are significant, which the authors should be more transparent about. For instance, in line 156-158 the associations are actually very low ($r=0.07-0.1$)

3. Data from a lot of different cohorts are included and it is a bit hard to follow what cohort has been used where. For instance, in the method section under “Analysis of Structured MRI data” PNC, HCP development, and HCP youth are listed as included neuroimaging data. There is very limited information, or no information, on what these cohorts are, if the data has been quality checked, or how many have been excluded. For PNC there is also no information on preprocessing if this has been done.

4. The introduction of the paper is quite focused on adolescence, therefore the switch to UKBIO came as a bit of a surprise. It is also unclear to me how some of the claims can be made given the limitations of the study.

For instance, in the abstract this is stated: "Group 3 showed increasing GMV and delayed neurocognitive development during adolescence due to a genetic origin, while these disadvantages were attenuated in mid-to-late adulthood"

Firstly, it is unclear to me how this paper provides adequate evidence that the pattern in group 3 has a purely genetic origin. Secondly, I do not see how the claim that the disadvantages were attenuated in mid-to-late adulthood can be made if the authors are here referring to the findings from UKBIO. Even if somebody has the genetic variants that were found to be associated with a proxy for group-3 developmental pattern in the ABCD, it does not necessarily mean that they actually had this pattern growing-up.

Other comments:

4. To me it was a surprise that the authors chose to use gray matter volume as a measure of brain morphometrics. Several studies have demonstrated that thickness and surface area are more accurate measures than cortical GMV. This is mainly because GMV is estimated based on the thickness and area of the cortex, but these two measures have been found to be associated with different genetic variants. Given that, what is the rationale for choosing GMV over cortical thickness or surface area, since both are readily available since the data was processed through freesurfer?

5. The authors state that scan site is used as a covariate. In for instance the ABCD study there are several sites with more than one scanner, so it would be more accurate to control for scanner. In the field there is also a lot of discussion on how to best control for the effects of scanner in multi-scanner imaging studies. Though it might give similar results to more advanced methods, the authors have here gone for a "simpler" method of doing this. What is the rationale for using this approach, and not other methods such as ComBat? And can the authors provide some evidence that they were able to factor out this effect in their

analysis?

6. The authors have used sex as a covariate. The pattern of development of gray matter has slightly different pattern for girls and boys, especially in pre-adolescent/early-adolescence. Within the brain charting field it is therefore quite common to estimate trajectories for females and males separately (see for instance Bethlehem et al., 2023). Therefore, I am curious to why the authors did not stratify the analysis by sex?

7. Several of the ABCD studies I am familiar with have controlled for family relations when working on the genetics data (see for instance Hughes et al., 2023). The ABCD study is oversampled for siblings and twins, and thereby has a nested structure, which should be considered. I might have missed it, but I cannot see that this has been done by the authors.

8. From the paper it appeared that they checked for associations between the clusters of GMV development, ADHD and Depression symptoms. It is unclear to me the rationale for only examining these mental health symptoms.

9. In the abstract the authors have written "In summary, our study revealed novel clusters of adolescent structural neurodevelopment and highlighted its long-term impacts on mental well-being and socio-economic outcomes".

This could be made clearer. Given the design of this paper, I would assume that long-term impact here refers to the analysis done on UKBIO. The way I understood the paper, the authors did not find any associations here.

10. In general, there is some inconsistency in terms of the use of abbreviations. There are also some abbreviations that are used that are never written out.

11. In the method section there are several aspects of the analysis where information on software and package usage is not listed, while for instance for the mediation analysis both the package and function used is not listed.

Point-by-Point Response to the Reviewers' Comments

Runye Shi, Shitong Xiang, Tianye Jia, Trevor W. Robbins, ..., Gunter Schumann, Xiaolei Lin*, Barbara J. Sahakian*, Jianfeng Feng*, IMAGEN Consortium

Enclosed, please find the revised submission of the paper "Structural neurodevelopment at the individual level - a life-course investigation using ABCD, IMAGEN and UK Biobank data", for publication in Nature Communications. We are thankful to the reviewers and their useful comments. We have fully addressed them. Below we provide the point-by-point response to both reviewers.

Reviewer #1 (Remarks to the Author):

Shi et al analysed sMRI data from the IMAGEN cohort and found evidence that adolescents can be clustered into three groups. This finding supports and extends previous findings of marked heterogeneity in structural neurodevelopment (e.g., ref 17-19). In addition, they used data from the ABCD, IMAGEN and UKB to study factors that could contribute to the heterogeneity, and also to predict long-term impacts in adulthood.

Overall, the team has produced an impressive set of results, and I fully support their emphasis on individual heterogeneity. Their efforts to combine and integrate various databases were creative, but as detailed below, it is also potentially a major weakness of the report.

1) The Introduction is well written, but presumably due to space limitations several crucial aspects of the report are not introduced. These include the use of the Cambridge Gambling Task to substantiate a differentiation between PC1 and PC2, and later on the use of ABCD and UKB cohorts in analyses of (epi)genetics and long-term outcomes.

Response: Thanks for the comments. We did not include the details regarding CGT and description of all three population cohorts due to space limitation. A more detailed description about the preprocessing of imaging and (epi)genetic data, instruments for neurocognitive assessments in ABCD, IMAGEN, UKB, and population cohorts (ABCD, IMAGEN, UKB, HCP and PNC) used in this study was provided in the **Supplementary Methods**.

2) The authors stress the use of a data-driven approach, but the analyses of developmental trajectories were based on 34 cortical and 10 subcortical ROIs. Was it for computational reasons that the analysis was limited to certain ROIs? This could be clarified and may warrant a comment that the analysis was not purely data-driven but rested on select ROIs. This issue is underscored by the use of PCA to identify a subset of ROIs.

Response: Thanks for pointing out this question. Since IMAGEN is a cohort of healthy adolescents, the change of brain ventricle system largely depends on the change of brain parenchyma. Therefore, here we only considered regional brain parenchyma in the identification of distinct neurodevelopment patterns. If we additionally add trajectories of lateral ventricle, inferior lateral ventricle, the 3rd ventricle, the 4th ventricle, the 5th ventricle, cerebrospinal fluid, vessel and choroid

plexus in the clustering, results remain similar (Table R1). Moreover, differences of the longitudinal trajectories in brain ventricle system were small among the three groups of adolescents (Table R2).

Table R1. Participants in each cluster overlapped between original analyses using 44 ROIs and analyses including additional brain ventricle systems.

Analyses including both brain ventricle and vessel systems	Original analyses		
	Group 1 (n=711)	Group 2 (n=765)	Group 3 (n=67)
Group 1 (n=675)	586	89	0
Group 2 (n=801)	125	671	5
Group 3 (n=67)	0	5	62

Table R2. Comparison of longitudinal trajectories in brain ventricle system among three groups of adolescents.

	Group 2 vs Group 1		Group 3 vs Group 1		Group 3 vs Group 2	
	Cohen's d	P _{adj}	Cohen's d	P _{adj}	Cohen's d	P _{adj}
Lateral ventricle	-0.27	1.11E-5	-0.19	1.00	-0.29	0.10
Inferior lateral ventricle	-0.25	9.16E-6	0.09	1.00	-0.14	1.00
The 3 rd ventricle	-0.34	1.40E-9	-0.59	3.63E-3	-0.79	1.01E-6
The 4 th ventricle	-0.28	4.84E-7	-0.09	1.00	-0.36	0.11
The 5 th ventricle	0.05	1.00	0.07	1.00	0.21	1.00
Cerebrospinal fluid	-0.20	1.18E-3	-0.71	0.02	-0.73	1.41E-3
Vessel	-0.12	0.102	0.12	1.00	-0.01	1.00
Choroid plexus	-0.25	8.67E-6	0.03	1.00	-0.22	0.33

3) Again, it was unclear why this dimension reduction was necessary given the data-driven emphasis. Finally, in subsequent analyses (line 185 ->), the authors again refer to certain ROIs, which was quite confusing in view of the initial analytic reduction steps.

Response: Thanks for the question. We apologize for the confusion and will explain why the dimension reduction was necessary below. Firstly, inter-individual heterogeneity of structural brain development was described using both developmental trajectories of total GMV and GMV trajectories in each ROI. Total GMV represents simple aggregation of gray matter volumes across all brain regions and may ignore the complex structural connections among different brain regions. Therefore, it is important to consider GMV trajectories in individual ROIs to identify potential population clustering patterns. However, current brain parcellation (using well-known established templates, such as Desikan-Killiany) may represent people's existing knowledge and understanding of the anatomically isolated and functionally specialized brain area, a more in-depth organization and functionally integration of the complicated neural system need to be considered. Here, by integrating regional GMV trajectories into principal components, we are investigating linear combinations of these brain regions, or, in other words, projection of the brain regions in a new linear space. Secondly, the multivariate k-means clustering method is subject to curse of dimensionality and would provide inaccurate clustering results when using lots of inter-correlated brain regions (Figure R1), since correlated variables carry extra weights on the distance calculation.

To answer your second question, we referred to certain ROIs in the subsequent analysis since we would like to understand intuitively which brain regions differed substantially among the three groups of adolescents. Because principal components, which were used in clustering of adolescents, involve linear combinations of ROIs and have limited interpretability in understanding the contribution of specific ROI, referring to certain ROIs as an ad-hoc analyses would be helpful in understanding the neurobiological basis of the group clustering.

Figure R1. Correlation matrix of regional GMV trajectories.

4) The first 15 PCs explained about 80% of the variance. Is it correct that the authors then restricted their clustering to PC1 and PC2? If so, how much variance did these two PCs account for? Is it possible that additional PCs, perhaps uncorrelated with baseline total GMV, would yield different results?

Response: Thanks for the question and we apologize for the confusion. The first 15 principal components capturing enough variance were used for population clustering, which was mentioned in lines 443-445:

"The first 15 principal component, which explained 80% of the total variation, were used in the following multivariate k-means clustering"

Additional PCs yield the same clustering results as the current results, but would introduce technical difficulties due to curse of dimensionality. It is generally a rule of thumb to include the smallest number of principal components as long as their cumulative variance contribution exceeds 75%.

5) The identification of distinct clusters was based on k-means clustering and the "elbow method". Cluster 1 (N=711) and 2 (N=765) were large, whereas cluster 3 only included 67 individuals (4.3%). The authors describe the difference between the first two clusters as quantitative in nature, with lower baseline total GMV and slower rate of GMV decrease in cluster 2. Thus, in a categorical sense, cluster 3 seems to be the deviating group, and it had higher PRS for ADHD than groups 1 and 2.

However, as mentioned, group 3 was a very small group and the analyses of genetic variants and epigenetic changes were therefore conducted on the ABCD cohort. Specifically, using "group-reweighted" GMV as proxy phenotype, due to the fact that the ABCD has a limited age-range, the authors assessed "one's tendency of being in Group 3 relative to Groups 1 and 2" (lines 239-241). This step was quite unclear to me. If I got it correctly, it will be based on baseline GMV regression weights from the relevant contrasts (2-1, 3-1, 3-2) and the top 10 ROIs with the largest loadings formed the basis for the group-reweighted GMV in ABCD (aged 9 years). While this was a creative way of bridging study cohorts, I found it to rest on many assumptions which undermines the strength of conclusions from the genetic and epigenetic analyses. The same critique holds for the extension to UKB.

Response: Thanks for the comments and we fully agree that the bridging from IMAGEN to ABCD are subject to several assumptions.

Specifically, the calculation of group-reweighted GMV in ABCD cohort relies on the assumption that brain regions exhibit comparable contrasts weights in IMAGEN and ABCD, and participants from IMAGEN and ABCD are drawn from a homogeneous population. In order for these assumptions to hold (or as good as it can hold), we considered linear GMV change from 9y (baseline age for majority of participants in ABCD) to 14y (baseline age for majority of participants in IMAGEN), which we believe is testable given the linear trend of GMV between 9y and 14y in brain chart¹ (Figure R2). To ensure participants from IMAGEN and ABCD were homogeneous, we selected only participants with self-reported ancestral origin "White" in ABCD. Given the fact that most of the participants in IMAGEN and ABCD were from middle-class communities and had relatively similar socio-economic status, we believe this homogeneous population assumption also makes sense.

Figure R2. Lifespan population trajectories of gray matter volume in males and females

Further, GWAS results (conducted in ABCD) were validated in IMAGEN, where group 3 showed higher PRS of late brain development compared to groups 1/2, and this PRS also showed positive correlation with improvement of neurocognition.

Given all these efforts, we believe that the bridging from IMAGEN to ABCD represent a worthwhile and reasonable try when direct connection of these two cohorts did not exist. We have included an explanation of these assumptions in lines 553-562 under the Methods section:

*"Since it was difficult to estimate individual GMV developmental trajectory in ABCD with limited number of structural MRI scans per participant and limited age range, we calculated the group-reweighted GMV as a proxy phenotype. **There are several underlying assumptions in this calculation. Firstly, it assumes that all brain regions exhibit a comparable linear change from childhood to adolescence. Secondly, it assumes that the participants from ABCD and IMAGEN are drawn from a homogeneous population. Once again, we only included individuals in ABCD with self-report White ancestral origins.** ROI-specific loading contributing to group classification (Group 2 vs Group 1, Group 3 vs Group 1, Group 3 vs Group 2, and Group 3 vs Groups 1/2) were obtained by regressing **baseline** GMV in 44 ROIs adjusting for age, sex, handedness and site in IMAGEN..."*

With regard to the bridging from IMAGEN to UKB, we could not do the group-reweighted kind of work since time lag between participants in these two cohorts are huge and it's challenging to ascertain the true effects of brain development in mid-to-late adulthood with all environmental influences entangled. Therefore, we only looked at the genetic predicted brain development, by looking at the effects of polygenic risk scores associated with delayed brain development. This is similar to the idea of Mendelian randomization, where genetic instruments (SNPs) associated with the exposure variable are used as the unconfounded proxies of the exposure variable. We have modified the manuscript to make this clear in lines 608-621 under the Methods section:

*"To estimate the long-term effect of delayed neurodevelopment, we calculated CENPW score and RPS according to the results of Group 3 GWAS and correlated these scores with outcomes of interest after regressing out the age effect at recruitment, site and gender. **It should be noted that these scores only reflect a genetic predicted risk for delayed brain development. Given the large age gap between participants in UK Biobank and IMAGEN, it is challenging to disentangle the long-term impacts of neurodevelopment from those due to potential environmental confounding in mid-to-late adulthood. Therefore, this analysis only serves to explore the potential long-term influence of genetically predicted delayed neurodevelopment and does not account for potential confounding due to environmental factors. Similarly, we assume the homogeneity of study populations between IMAGEN and UK Biobank.** For PRS calculation, we used P value thresholds from 5E-08 to 1 with a step of 5E-05 and calculated an average PRS score for each individual. Due to the large sample size and easily-obtainable statistical significance, inferiority tests against 0.05 were conducted against the null hypothesis that the absolute correlation coefficient was less than 0.05."*

6) Taken together, the authors mention in their discussion that the attempt to link the neurodevelopmental patterns from IMAGEN to ABCD and UKB "may subject to confounding bias". I fully agree and to me the paper might have come across as stronger had it involved the IMAGEN findings only.

Response: Thanks for the comment. We agree that there exist strong assumptions when bridging IMAGEN to ABCD and UKB, as mentioned in our response to your previous question. However, we believe that the analytical approach to connect large population cohorts, when used appropriately, is of great importance when long-term follow-up of IMAGEN is not yet available and when genetic analyses in a smaller cohort are underpowered. In addition, although findings from IMAGEN may seem to be enough for a complete original research article, we believe that a full picture of the heterogeneity with respect to adolescence neurodevelopment will provide a more pragmatic perspective for adolescent development as a whole.

7) A final comment. The authors conclude that the adolescents in group 3 are in critical need of support and counselling for their brain and neurocognitive abilities to better develop. Given that some differences diminished over time, I found this statement unclear.

Response: Thanks for the comment. Besides diminished differences in neurocognition between group 3 and group 1, we also observed increased depression symptoms in group 3. A previous study has found that childhood neurodevelopmental difficulties such as ADHD increased the risk for early onset depression in adolescence². They suggest that neurodevelopmental difficulties may lead to depression-related environmental exposures such as repeated experience of academic failure and peer rejection. Thus, school and family support were also needed to help prevent such emotional disturbance. In addition, as IMAGEN participants were recruited from middle-class communities, the improvement of neurocognition in group 3 was likely attributed to the fact that they already had enough family support. However, we are not sure whether this would be naturally occurring without appropriate interventions in the whole population.

Reviewer #2 (Remarks to the Author):

In this study the authors have clustered youth into three groups based on their developmental pattern of Grey Matter Volume. They looked for associations between these clusters, neurocognition and mental health symptoms. Next, they examined the genetic and epigenetic associations with each cluster.

All-in-all, Lin and colleagues present an interesting and impressive manuscript. I have several comments that I hope will help improve the manuscript. My first list of comments are mainly aspects of the manuscript, and the reasoning behind it, that I believe needs to be made clearer so that it will be easier for the reader to follow. I am not familiar with EWAS, so I have not made comments on that part of the analysis.

1) I am having difficulties with understanding how the group-reweighted GMV can be derived in the ABCD given that the participants here are outside of the age-range in the IMAGEN dataset. From my understanding this was solved by adding additional cohorts to the IMAGEN sample, and then predicting developmental trajectories of gray matter volume. But these cohorts that were added were then cross-sectional? So the “developmental trajectories” estimation which is used later as a proxy is then based on cross-sectional data and not longitudinal data?

Response: Thanks for your question and we apologize for the confusion. We have now explained this calculation in more details in lines 559-562:

*"ROI-specific loading contributing to group classification (Group 2 vs Group 1, Group 3 vs Group 1, Group 3 vs Group 2, and Group 3 vs Groups 1/2) were obtained by regressing **baseline GMV** in 44 ROIs adjusting for age, sex, handedness and site in IMAGEN. Logistic regression model was used as the classification model and top 10 ROIs with the largest loadings were used to calculate the group-reweighted GMV in ABCD."*

Specifically, this involves a two-step calculation. First, group labels (group 3 vs groups 1/2) identified from longitudinal change of GMVs were mapped to baseline GMVs, where the contribution of each ROI in contrasting group 3 vs groups 1/2 can be obtained from IMAGEN. Next, baseline GMVs in ABCD were adjusted for age (to be age consistent with IMAGEN) and ROI-specific loading calculated from IMAGEN can be applied to ABCD. Here, we assume that all brain regions exhibit a comparable linear change from childhood to adolescence¹ (Figure R2). This assumption was also added to lines 553-562:

*"Since it was difficult to estimate individual GMV developmental trajectory in ABCD with limited number of structural MRI scans per participant and limited age range, we calculated the group-reweighted GMV as a proxy phenotype. **There are several underlying assumptions in this calculation. Firstly, it assumes that all brain regions exhibit a comparable linear change from childhood to adolescence. Secondly, it assumes that the participants from ABCD and IMAGEN are drawn from a homogeneous population. Once again, we only included individuals in ABCD who self-report their ancestral origins as white.** ROI-specific loading contributing to group classification (Group 2 vs Group 1, Group 3 vs*

Group 1, Group 3 vs Group 2, and Group 3 vs Groups 1/2) were obtained by regressing **baseline GMV in 44 ROIs adjusting for age, sex, handedness and site in IMAGEN...**"

Figure R2. Lifespan population trajectories of gray matter volume in males and females

2) I also see that here there are 19 587 participants included from ABCD-study in making the age-specific GMV curves. To my knowledge there are not that many participants in the study, so have the authors then included multiple scans from the same participants? If this was the case, did you in any way control for the same person being included several times?

Response: Thanks for the question and we apologize for the confusion. The number 19,587 indeed referred to the number of scans. We have revised the Method section lines 460-465.

*"To illustrate the region-specific GMV development in an extended time frame ranging from late childhood to early adulthood, external neuroimaging data from several population cohorts were incorporated. This includes a total of 21,826 participants comprising of **11,811 participants aged 9-14y with 19,587 scans** in ABCD, 652 participants aged 5-22y in HCP-D, and 1,587 participants aged 8-23y in PNC study."*

Here we used linear mixed effect model with random intercept and slope at the subject level to account for the within-subject correlation.

3) In the result section the strengths of the associations are not given in the text but referred to in the supplement. Some of the associations that are reported have a very low effect size even though they are significant, which the authors should be more transparent about. For instance, in line 156-158 the associations are actually very low ($r=0.07-0.1$)

Response: Thanks for the suggestion. We agree that both effect sizes and statistical significance should be included in the manuscript so that readers would understand the results and conclusions better. However, small effect sizes are expected in this study since the population cohort in the analyses consists of healthy individuals (rather than patients versus healthy controls). We have revised the Results section lines 158-162.

"PC1 was significantly associated with delay aversion ($r = 0.07$, $P_{adj} = 0.30$) and risk adjustment ($r = -0.08$, $P_{adj} = 0.020$), and PC2 was significantly associated with deliberation time ($r = 0.1$,

$P_{adj} = 0.003$), overall betting (proportion bet) ($r = 0.07$, $P_{adj} = 0.014$) and risk-taking ($r = 0.08$, $P_{adj} = 0.008$)."

Due to space limitations, it was hard to report all P values when comparing neurocognition performances and mental health symptoms between Group 2, Group 3 and Group 1 (the reference Group). However, all test statistics were available in the Supplementary Table 4 and some were reported in Fig. 2 of the manuscript, where most of the effect sizes were larger than 0.1.

4) Data from a lot of different cohorts are included and it is a bit hard to follow what cohort has been used where. For instance, in the method section under "Analysis of Structured MRI data" PNC, HCP development, and HCP youth are listed as included neuroimaging data. There is very limited information, or no information, on what these cohorts are, if the data has been quality checked, or how many have been excluded. For PNC there is also no information on preprocessing if this has been done.

Response: Thanks for the suggestion and we apologize for the confusion. A full description of the use of population cohorts and related data preprocessing and quality check workflow has been included in the **Supplement Methods**. In summary, data from PNC and HCP were only used in deriving the population GMV trend during childhood and adolescence, while other analyses involved in this paper used IMAGEN, ABCD and UKB. The primary analyses investigating population clustering of brain developmental trajectories were carried out in IMAGEN, while genetic analyses were performed in ABCD since IMAGEN is underpowered due to limited sample size.

5) The introduction of the paper is quite focused on adolescence, therefore the switch to UKBIO came as a bit of a surprise. It is also unclear to me how some of the claims can be made given the limitations of the study.

For instance, in the abstract this is stated: "Group 3 showed increasing GMV and delayed neurocognitive development during adolescence due to a genetic origin, while these disadvantages were attenuated in mid-to-late adulthood"

Firstly, it is unclear to me how this paper provides adequate evidence that the pattern in group 3 has a purely genetic origin. Secondly, I do not see how the claim that the disadvantages were attenuated in mid-to-late adulthood can be made if the authors are here referring to the findings from UKBIO. Even if somebody has the genetic variants that were found to be associated with a proxy for group-3 developmental pattern in the ABCD, it does not necessarily mean that they actually had this pattern growing-up.

Response: Thanks for the comments. We apologize for the confusion. However, we did not mean that group 3 has a purely genetic origin. The evidence found in this analysis support that the difference between group 3 and group 1 had a genetic explanation, while differences between group 2 and group 1 (seem like a quantitative difference) did not. We realized that readers may misinterpret our findings and therefore, we added this as a limitation in the Discussion section.

"By integrating genomic, neuroimaging, behavior and health-related data from three large-scale population cohorts, we confirmed that genetic variants are associated with delayed brain maturation and neurocognitive development, without affecting the socio-economic and mental well-being later in life. Whereas, adverse environmental exposure and the associated epigenetic changes could lead to prolonged negative effects on brain development and behavioral disadvantages...***However, it does not necessarily mean that the differences between Group 3 and Group 1 could only be attributed to genetic variation, or that differences between Group 2 and Group 1 was purely due to environment. Future research with larger sample size and adequate statistical power are needed to elucidate the potential interplay between gene and environment on structural brain development.***"

For your second concern, we appreciate this question as the other reviewer also pointed out similar critiques. By investigating the effect of polygenic risk score of delayed brain development in UK Biobank participants, we are actually referring to the genetic proxies of delayed brain development, rather than the true phenotype of delayed brain development. It would be challenging to disentangle the effects of structural brain development and confounding's from potential environmental factors due to the large age gap between participants in IMAGEN and UK Biobank. The idea behind polygenic risk score was similar to Mendelian randomization, where SNPs associated with the exposure (at genome-wide significance level) were treated as proxies to the exposure, and association were conducted between SNPs and outcomes of interest. To avoid misinterpretation, we have added to the Method section in lines 608-621.

" To estimate the long-term effect of delayed neurodevelopment, we calculated CENPW score and RPS according to the results of Group 3 GWAS and correlated these scores with outcomes of interest after regressing out the age effect at recruitment, site and gender. It should be noted that these scores only reflect a genetic predicted risk for delayed brain development. Given the large age gap between participants in UK Biobank and IMAGEN, it is challenging to disentangle the long-term impacts of neurodevelopment from those due to potential environmental confounding in mid-to-late adulthood. Therefore, this analysis only serves to explore the potential long-term influence of genetically predicted delayed neurodevelopment and does not account for potential confounding due to environmental factors. Similarly, we assume the homogeneity of study populations between IMAGEN and UK Biobank. For PRS calculation, we used P value thresholds from 5E-08 to 1 with a step of 5E-05 and calculated an average PRS score for each individual. Due to the large sample size and easily-obtainable statistical significance, inferiority tests against 0.05 were conducted against the null hypothesis that the absolute correlation coefficient was less than 0.05."

Other comments:

6) To me it was a surprise that the authors chose to use gray matter volume as a measure of brain morphometrics. Several studies have demonstrated that thickness and surface area are more accurate measures than cortical GMV. This is mainly because GMV is estimated based on the thickness and area of the cortex, but these two measures have been found to be associated with different genetic variants. Given that, what is the rationale for choosing GMV over cortical thickness or surface area, since both are readily available since the data was processed through freesurfer?

Response: Thanks for your question. A remarkable characteristic of brain development in adolescence is an imbalance between the more mature subcortical areas and less mature prefrontal areas^{3,4}. Therefore, we consider it necessary to include both the cortical and subcortical brain regions when dealing with adolescent brain development, rather than only cortical thickness or surface area. It may seem more reasonable to consider cortical thickness or surface area in imaging genetics studies, additionally with the shape or volume of subcortical regions. Comprehensive analyses with consistent results using multiple brain measures may help better integrate and interpret the results. Clustering of adolescence using different brain measures were provided in Table R3 and R4 below. Considerable overlap was observed among these different measures in terms of Groups 1 and 2. However, group 3, which was characterized by opposite directions of GMV growth trajectories, could not be identified or characterized comprehensively using other brain measures. Therefore, using only cortical surface area and thickness as grouping measures may not be sufficient to capture distinct neurodevelopment patterns for adolescents.

Table R3. Group overlap between clustering using GMV and cortical thickness.

Groups identified using cortical thickness	Groups identified using GMV		
	Group 1 (n=711)	Group 2 (n=765)	Group 3 (n=67)
Group 1 (n=531)	446	81	2
Group 2 (n=808)	257	547	0
Group 3 (n=219)	8	137	65

Table R4. Group overlap between clustering using GMV and cortical surface area.

Groups identified using surface area	Groups identified using GMV		
	Group 1 (n=711)	Group 2 (n=765)	Group 3 (n=67)
Group 1 (n=1389)	710	672	0
Group 2 (n=139)	1	93	43
Group 3 (n=30)	0	0	24

7) The authors state that scan site is used as a covariate. In for instance the ABCD study there are several sites with more than one scanner, so it would be more accurate to control for scanner. In the field there is also a lot of discussion on how to best control for the effects of scanner in multi-scanner imaging studies. Though it might give similar results to more advanced methods, the authors have here gone for a "simpler" method of doing this. What is the rationale for using this approach, and not other methods such as ComBat? And can the authors provide some evidence that they were able to factor out this effect in their analysis?

Response: Thanks so much for this suggestion. Please find our detailed replies as follows:

(1) In the original analyses, we only included site as a covariate and did not consider scanner (or, frankly speaking, not aware of the scanner effect). Following your suggestion, here we reanalyzed the ABCD data using scanner (*mri_info_deviceserialnumber* variable in *abcd_mri01.txt*) as covariates to adjust for batch effect. The adjustment of scanners was emphasized in the Supplementary Methods:

*"The quality control (QC) procedure of the processed neuroimages was checked by the ABCD team both automatically and manually. Then regional morphometric structure evaluations were obtained using FreeSurfer 6.0 including cortical volumes from the FreeSurfer Desikan-Killiany (h.aparc) atlas, and subcortical volumes from the ASEG atlas. According to the FreeSurfer reconstruction QC measures (freesqc01), a total of 1,9576 scans including 11,811 participants passed the QC were included in the structural analyses. **As there is more than one scanner in several sites, we controlled for scanner (mri_info_deviceserialnumber variable in abcd_mri01 file) rather than site in the analysis. A total of 31 scanners was used in ABCD. Because most of the sample size of each scanner was large enough, regression method is able to substantially mitigate scanner effects.**"*

Meanwhile, considered sibling effect according to your suggestion #9 in GWAS, results for the 'GWAS and validation' part, 'long-term impacts of neurodevelopment in UK Biobank' part have been updated, which did not change much compared to the original analyses, except that SNPs on *ADGRL3* identified in the previous results didn't pass the genome-wide significance level in the updated analyses (Figure R3).

Figure R3. A comparison of results in the ‘GWAS and validation’ part, and ‘long-term impacts of neurodevelopment in UK Biobank’ part between the previous analysis (considering between-site variation) and current analysis (considering between-scanner variation and sibling effect). The referred figure number was added behind the subtitle, and Supplementary Figure 12 in the previous results was changed to Supplementary Figure 11 in the updated results.

(2) We fully agree that methods to control for batch effects would yield similar results when using regression approaches and ComBAT. There are two reasons why we chose regression approach to control for the batch effect instead of ComBAT. Firstly, an advantage of ComBAT over other methods including L/S batch adjustment is that it is more robust to outliers in small sample sizes⁵. However, most of the sample sizes within each scanner were large in ABCD (Table R5; using baseline scan as an example). Secondly, in ComBAT, an empirical Bayes method was used to shrink the parameter estimates where prior distributions of the batch parameters were obtained from L/S model. Results from ComBAT usually agree with the regression approach given adequate sample sizes. Figure R3 and R4 showed that both regression approach and ComBAT are able to substantially mitigate batch effects in ABCD, and their results converge.

Table R5. The distribution of participants across 29 scanners in the ABCD baseline data.

	Male (n)	Female (n)	Site
scanner01	229	218	site11
scanner02	274	236	site14
scanner03	296	255	site02
scanner04	186	192	site05
scanner05	24	21	site12
scanner06	560	451	site16
scanner07	14	22	site22
scanner08	224	235	site04
scanner09	221	209	site09
scanner10	264	272	site19
scanner11	334	296	site03
scanner12	181	166	site08
scanner13	184	154	site07
scanner14	95	83	site13
scanner15	201	196	site01
scanner16	234	201	site15
scanner17	48	41	site14
scanner18	293	289	site06
scanner19	204	177	site18
scanner20	314	257	site21
scanner21	276	269	site13
scanner22	115	113	site20
scanner23	236	229	site20
scanner24	304	270	site10
scanner25	299	279	site17
scanner26	78	85	site10
scanner27	285	264	site12
scanner28	10	10	site21
scanner29	163	12	site04

Figure R4. Total gray matter volume before and after normalizing by regression (Linear) and ComBat from the ABCD baseline. The left column shows raw volumetric data across 29 scanners included in ABCD baseline, the middle column shows normalized data by regression and the bottom row shows data normalized using ComBat. ANOVA P-values refer to one-way analyses of variance across sites.

Figure R5. Comparing effects of ComBat versus regression (Linear) batch correction on the estimation of total gray matter volume.

8) The authors have used sex as a covariate. The pattern of development of gray matter has slightly different pattern for girls and boys, especially in pre-adolescent/early-adolescence. Within the brain charting field it is therefore quite common to estimate trajectories for females and males separately (see for instance Bethlehem et al., 2023). Therefore, I am curious to why the authors did not stratify the analysis by sex?

Response: Thanks for the comment. We agree that sex differences did exist and was important in identifying GMV growth trajectories. Therefore, we updated the analyses by conducting the analyses separately for males and females. In summary, clustering results and analysis of neurocognition/mental health symptoms largely overlapped with the original analyses (**Supplementary Table 7**), but we did find some interesting results: difference of neurocognition among subgroups was manifested more for risk-taking behaviors and impulsion in males, while for spatial working memory in females (new analyses results provided in **Supplementary Table 8**); increase of the depressive symptoms in Group 2 was only observed among males, and increase of depressive symptoms in Group 3 was observed only among females. These results were included in the second part of the results section lines 228-236:

" Given the slightly different patterns of GMV development for males and females²⁸, we conducted the analyses stratified by sex following the same workflow. Results of group clustering largely overlapped with the original analyses (Supplementary Table 7). In general, the sex-stratified analyses revealed similar patterns of neurocognition and mental health symptoms among three groups of adolescents. However, differences of neurocognition among these groups were manifested more for risk-taking and impulsive behaviors in males, while for spatial working memory in females (Supplementary Table 8). Besides, increase of the depressive symptoms in Group 2 was only observed in males, and increase of the depressive symptoms in Group 3 was only observed in females."

New Supplementary Table 7. The overlapping number of participants between clustering results within different sex and the one within the whole population

Clustering results within the whole population			
Clustering results within males			
	Group 1 (n=376)	Group 2 (n=332)	Group 3 (n=39)
Group 1 (n=380)	374	6	0
Group 2 (n=335)	2	326	7
Group 3 (n=32)	0	0	32
Clustering results within females			
	Group 1 (n=335)	Group 2 (n=433)	Group 3 (n=28)
Group 1 (n=333)	327	6	0
Group 2 (n=425)	8	417	0
Group 3 (n=38)	0	10	28

A short version of new Supplementary Table 8. Full comparison of Group 2 and Group 3 vs Group 1 clustered within sex in terms of the personal traits, environmental burden, neuro-cognition, behavioral risk factors and mental symptoms at age 14, longitudinal trajectory and at age 23.

Question naire	Item	Time	Clustering results within males						Clustering results within females					
			Group 2 vs Group 1		Group 3 vs Group 1		Group 3 vs Group 2		Group 2 vs Group 1		Group 3 vs Group 1		Group 3 vs Group 2	
			d	P _{adj}	d	P _{adj}	d	P _{adj}	d	P _{adj}	d	P _{adj}	d	P _{adj}
PRM	Percent correct	14y	-0.09	0.648	0.23	0.254	0.32	0.164	0.03	0.506	-0.33	0.034	-0.37	0.013
		trajectory	0.04	0.824	-0.87	0.009	-0.82	0.007	-0.01	0.943	0.08	0.739	0.08	0.699
		19y	-0.02	0.722	-0.58	0.119	-0.37	0.148	-0.07	0.344	-0.20	0.273	-0.13	0.487
AGN	Total omissions for positive category	14y	0.08	0.138	-0.03	0.915	-0.12	0.816	-0.08	0.914	0.24	0.036	0.33	0.034
		trajectory	-0.09	0.290	-0.11	0.559	-0.01	0.887	-0.03	0.744	-0.19	0.408	-0.18	0.485
		19y	0.03	0.896	-0.14	0.627	-0.17	0.651	-0.02	0.855	0.28	0.427	0.38	0.344
	Total omissions for negative category	14y	0.07	0.138	0.10	0.795	0.04	0.816	-0.08	0.914	0.31	0.024	0.41	0.013
		trajectory	-0.14	0.186	-0.34	0.213	-0.20	0.686	-0.09	0.592	-0.50	0.059	-0.45	0.158
		19y	-0.01	0.896	-0.10	0.627	-0.12	0.651	-0.04	0.855	0.17	0.462	0.27	0.351
SWM	Between error	14y	-0.07	0.866	0.22	0.432	0.31	0.379	-0.04	0.364	0.38	0.002	0.42	0.015
		trajectory	-0.01	0.783	-0.32	0.258	-0.30	0.295	0.11	0.177	-0.14	0.780	-0.27	0.346
		23y	-0.02	0.846	0.10	0.650	0.12	0.579	0.13	0.076	0.23	0.237	0.11	0.967
	Strategy	14y	0.01	0.719	0.11	0.498	0.11	0.781	-0.02	0.492	0.32	0.021	0.33	0.045

		trajectory	-0.10	0.402	-0.05	0.722	0.06	0.857	0.20	0.017	0.01	0.780	-0.20	0.346
		23y	-0.08	0.728	0.26	0.588	0.33	0.273	0.18	0.025	0.17	0.237	-0.01	0.967
CGT	Delay aversion	14y	0.05	0.893	-0.01	0.797	-0.07	0.962	-0.09	0.672	0.54	0.004	0.63	0.004
		trajectory	0.00	0.976	0.16	0.871	0.16	0.498	0.15	0.509	-0.04	0.993	-0.19	0.749
		23y	0.18	0.046	0.12	0.941	-0.07	0.828	0.11	0.196	0.18	0.539	0.06	0.830
	Deliberation time	14y	-0.21	0.145	0.07	0.210	0.36	0.038	0.03	0.672	0.03	0.615	0.00	0.885
		trajectory	0.23	0.052	-0.05	0.952	-0.36	0.304	0.03	0.896	-0.15	0.993	-0.12	0.749
		23y	-0.02	0.762	-0.10	0.941	-0.08	0.828	0.12	0.196	-0.22	0.639	-0.21	0.660
	Overall proportion bet	14y	-0.04	0.893	0.40	0.034	0.43	0.038	-0.02	0.672	0.08	0.265	0.10	0.401
		trajectory	0.20	0.055	-0.13	0.871	-0.33	0.251	-0.04	0.896	0.03	0.993	0.07	0.749
		23y	0.20	0.028	0.02	0.941	-0.18	0.828	0.01	0.891	0.14	0.539	0.13	0.718
	Quality of decision making	14y	0.00	0.931	-0.31	0.023	-0.34	0.022	0.05	0.672	-0.33	0.006	-0.39	0.011
		trajectory	-0.18	0.063	0.23	0.871	0.43	0.251	-0.08	0.896	0.17	0.993	0.25	0.749
		23y	-0.11	0.196	0.02	0.941	0.13	0.828	-0.12	0.196	-0.12	0.539	0.00	0.945
	Risk adjustment	14y	0.04	0.893	-0.47	0.018	-0.50	0.022	-0.02	0.672	-0.50	0.004	-0.46	0.011
		trajectory	-0.20	0.055	0.13	0.871	0.35	0.251	0.01	0.941	-0.06	0.993	-0.07	0.749
		23y	-0.12	0.146	-0.28	0.755	-0.18	0.828	-0.11	0.196	-0.46	0.072	-0.35	0.391
	Risk taking	14y	-0.03	0.893	0.34	0.036	0.35	0.063	-0.01	0.672	0.06	0.277	0.07	0.421
		trajectory	0.19	0.055	-0.19	0.871	-0.37	0.251	-0.04	0.896	0.06	0.993	0.10	0.749
		23y	0.22	0.028	-0.05	0.941	-0.28	0.828	0.01	0.891	0.14	0.539	0.14	0.718
DAWBA	Major Depression	14y	0.17	0.761	0.07	0.956	-0.10	0.771	-0.13	0.619	0.11	0.960	0.30	0.639
		trajectory	0.10	0.275	-0.10	0.881	-0.13	0.594	0.11	0.140	0.76	0.000	0.64	0.001
		23y	0.22	0.015	-0.14	0.975	-0.23	0.374	0.06	0.345	0.82	0.000	0.58	0.001
	ADHD (child)	14y	-0.05	0.036	0.31	0.435	0.40	0.082	0.00	0.351	0.24	0.019	0.23	0.068
		trajectory	0.07	0.457	-0.20	0.378	-0.31	0.231	0.03	0.781	-0.31	0.158	-0.33	0.118
		23y	-0.08	0.333	-0.01	0.882	0.08	0.824	0.10	0.165	-0.02	0.953	-0.11	0.581
	ADHD (parent)	14y	0.06	0.155	0.35	0.030	0.28	0.105	0.04	0.727	0.28	0.120	0.23	0.145
		16y	-0.01	0.982	0.19	0.328	0.23	0.310	-0.02	0.855	0.25	0.166	0.28	0.128

9) Several of the ABCD studies I am familiar with have controlled for family relations when working on the genetics data (see for instance Hughes et al., 2023). The ABCD study is oversampled for siblings and twins, and thereby has a nested structure, which should be considered. I might have missed it, but I cannot see that this has been done by the authors.

Response: Thank you for providing the suggestion and references. We agree that including twins or siblings would decrease the effective sample sizes in GWAS due to correlated genetic components. Therefore, we updated our analyses by randomly picking one participant out of the siblings or twins within each family (The kinship relationship between participants was obtained by genetically inferred zygosity status in *acspsw03* file). 1,052 participants were excluded from the GWAS. The description of this preprocessing process was included in the Methods section:

*"...In this study, we performed stringent QC standards using PLINK 1.90...For ABCD, we only selected subjects with self-reporting White ancestral origins using the public release 3.0 imputed genotype data, which was imputed with the HRC reference panel. **Considering that ABCD is oversampled for siblings and twins, we randomly selected one participant within each family...**"*

and Supplementary Methods.

*"The ABCD imputed genotype data were obtained from the public release 3.0. Imputation was performed using the Michigan Imputation Server with *hrc.r1.1.2016* reference panel and Eagle v2.3 phasing. We performed stringent QC standards by PLINK 1.90. Individuals with >10% missing rate and single-nucleotide polymorphisms (SNPs) with call rates < 95%, minor allele frequency < 0.1%, deviation from the Hardy-Weinberg equilibrium with $P < 1E-10$ were excluded from the analysis, yielding 11,1014 participants and 244,227 SNPs. To ensure the homogeneity of the ABCD and IMAGEN population, we selected only ABCD subjects self-reporting ancestral origins as white, with 2,387 participants excluded. **Considering that ABCD is oversampled for siblings and twins, and thereby has a nested structure, we randomly selected one participant within a family (the kinship relationship between participants was decided by genetically inferred zygosity status in *acspsw03* file).** Finally, a total of 7,662 participants was included in the genetic analysis."*

In addition, we also controlled for scanner effects according to your suggestion #7 in GWAS. Results have been updated, which did not change much compared to the original analyses, except that SNPs on *ADGRL3* didn't pass the genome-wide significance level in the updated analyses (Figure R3).

10) From the paper it appeared that they checked for associations between the clusters of GMV development, ADHD and Depression symptoms. It is unclear to me the rationale for only examining these mental health symptoms.

Response: Thanks for your question. As suggested by Dylan et al., genes associated with neurodevelopment diseases (ADHD, autism, early onset depression and Tourette syndrome) predicted psychiatric symptoms through early adolescence with great sensitivity⁶. However, as the study population of IMAGEN is a healthy population, the prevalence of Tourette syndrome at baseline was less than 0.01%. Besides, the follow-up rate of participants in IMAGEN with non-missing parent-rated autism scores was quite low. Therefore, we only considered ADHD and depression symptoms in comparing the risks for psychiatric disorders among three groups of adolescents.

11) In the abstract the authors have written “In summary, our study revealed novel clusters of adolescent structural neurodevelopment and highlighted its long-term impacts on mental well-being and socio-economic outcomes”. This could be made clearer. Given the design of this paper, I would assume that long-term impact here refers to the analysis done on UKBIO. The way I understood the paper, the authors did not find any associations here.

Response: Thanks for the suggestion. We agree that this expression may lead to misunderstanding. Therefore, we have modified the abstract lines 94-97 to:

*"In summary, our study revealed novel clusters of adolescent structural neurodevelopment and suggested that **genetically-predicted delayed neurodevelopment has limited long-term effects on mental well-being and socio-economic outcomes later in life.**"*

12) In general, there is some inconsistency in terms of the use of abbreviations. There are also some abbreviations that are used that are never written out.

Response: Thanks for the suggestion and we apologize for the inconsistency and confusion. We have clarified all the abbreviations in the revised manuscript and Figures. Specifically, we have added clarifications in the manuscript as shown below:

lines 86-89 in the Abstract,

*"Genetic and epigenetic determinants of group clustering and long-term impacts of neurodevelopment in mid-to-late adulthood were investigated using data from **the Adolescent Brain Cognitive Development (ABCD), IMAGEN and UK Biobank cohorts.**"*;

lines 218-220 in the Results section,

*"Consistent with the improvements of neurocognition, we observed decreased **attention-deficit/hyperactivity disorder (ADHD)** symptoms, but increased depression symptoms in Group 3..."*;

lines 411-412 in the Methods section,

*"Individuals with GMV beyond 4 **interquartile ranges (IQRs)** in any ROI were considered as outliers and were excluded from the analyses."*;

Figure 2,

CANTAB was specified as Cambridge Neuropsychological Test Automated Battery and ADHD was specified as attention-deficit/hyperactivity disorder;

Figure 3,

CGT was specified as Cambridge Gambling Task and SST GoRT was specified as reaction time for ‘Go’ trails in Stop Signal Task;

Figure 4,

Chldexp was specified as child's experience of family life, FamStress was specified as family stressors, and CTQ was specified as Childhood Trauma Questionnaire;

Figure 5,

IMD was specified as Indices of Multiple Deprivation, Edu was specified as the highest educational level, and IQ was specified as intelligence.

We also corrected the inconsistency of the use of abbreviations in Figure 3, as shown below.

Fig. 3 Genome-wide association study (GWAS) identified two significant loci associated with delayed neurodevelopment in Group 3.

(a) Correlation between Group3-reweighted GMV and neurocognition (Supplementary Methods) in ABCD ($n=11,101$) indicated the validity of using the proxy phenotype for delayed neurodevelopment in the following GWAS. GDT, Game of Dice Task; DDT, Dealy Discounting Task; PVT, Picture Vocabulary Test; Flanker, Flanker Inhibitory Control and Attention Test; List, List Sorting Working Memory Test; DCCS, Dimensional Change Card Sort Test; Pattern, Pattern Comparison Processing Speed Test; PSMT, Picture Sequence Memory Test; Reading, Oral Reading Recognition Test; FluidCog, fluid cognition; CrystalCog, crystallized cognition; TotalCog, total cognition, *** $P < 0.001$; ** $P < 0.01$; * $P < 0.05$. (b) GWAS Manhattan plot for Group3-reweighted GMV in the ABCD population ($n=7,662$). Group3-reweighted GMV was calculated for each adolescent (Methods) and used as the proxy phenotype for delayed neurodevelopment. Multiple SNPs on chromosome 6 achieved genome-wide significant effects ($P < 5 \times 10^{-8}$). SNPs on chromosome 6 were mapped to the intronic region of CENPW. Results from gene based association analysis (Supplementary Fig. 11) confirmed the significant effect of CENPW on delayed neurodevelopment. Box plot in (c) showed that CENPW score of delayed neurodevelopment was higher in Group 3 ($n=60$) compared to Group 1 and 2 ($n=1,338$) (two-sided t-test: $P=0.028$). The upper and lower boundaries of each boxplot represented the first (Q1) and third (Q3) quantiles, respectively. Hence, the box body covered 50% of the central data, with the median marked by a central line. The top/bottom whiskers represented the maximum or minimum, respectively without outliers. (d) indicated that CENPW score of delayed neurodevelopment was negatively correlated with baseline (BL) neurocognitive performance, and became non-significant at the last follow-up (FU). Here, Worse indicated higher CGT Delay aversion score, lower CGT risk adjustment score, longer CGT Deliberation time and SST GoRT. CGT Delay aversion, BL ($r=0.09$, * $P_{adj}=0.027$), FU3 ($r=0.07$, $P_{adj}=0.239$); CGT Deliberation time, BL ($r=0.08$, * $P_{adj}=0.027$), FU3 ($r=-0.02$, $P_{adj}=0.983$); CGT risk adjustment, BL ($r=-0.08$, * $P_{adj}=0.027$), FU3 ($r=0.022$, $P_{adj}=0.983$); SST GoRT, BL ($r=-0.06$, * $P_{adj}=0.038$), FU3 ($r=-0.03$, $P_{adj}=0.472$). CGT, Cambridge Gambling Task; SST GoRT, reaction time for 'Go' trials in Stop Signal Task. (c-d) confirmed the relationship between CENPW and delayed neurodevelopment identified in (b).

13) In the method section there are several aspects of the analysis where information on software and package usage is not listed, while for instance for the mediation analysis both the package and function used is not listed.

Response: Thanks for the suggestion. We have checked all relevant software and packages used in the analyses and provided the details information in the Methods section. A short description of software used in the analysis was also described in the Code Availability lines 635-639:

"Primary analyses were conducted in R v4.2.2. Linear mixed effect models were performed using lme4 1.1-31 and nlme 3.1-160 R packages. Mediation analysis was performed using lavaan 0.6-12 R package. PLINK 2.0 was used to perform GWAS and calculate CENPW score. MAGMA v1.10 was used to perform the gene-based association analysis. PRSice v2.3.3 was used to calculate the PRS."

References

- 1 Bethlehem, R. A. I. *et al.* Brain charts for the human lifespan. *Nature* **604**, 525-533, doi:10.1038/s41586-022-04554-y (2022).
- 2 Rice, F. *et al.* Characterizing Developmental Trajectories and the Role of Neuropsychiatric Genetic Risk Variants in Early-Onset Depression. *JAMA Psychiatry* **76**, 306-313, doi:10.1001/jamapsychiatry.2018.3338 (2019).
- 3 Konrad, K., Firk, C. & Uhlhaas, P. J. Brain development during adolescence: neuroscientific insights into this developmental period. *Dtsch Arztebl Int* **110**, 425-431, doi:10.3238/arztebl.2013.0425 (2013).
- 4 Casey, B. J., Jones, R. M. & Hare, T. A. The adolescent brain. *Ann N Y Acad Sci* **1124**, 111-126, doi:10.1196/annals.1440.010 (2008).
- 5 Johnson, W. E., Li, C. & Rabinovic, A. Adjusting batch effects in microarray expression data using empirical Bayes methods. *Biostatistics* **8**, 118-127, doi:10.1093/biostatistics/kxj037 (2007).
- 6 Hughes, D. E. *et al.* Genetic patterning for child psychopathology is distinct from that for adults and implicates fetal cerebellar development. *Nat Neurosci* **26**, 959-969, doi:10.1038/s41593-023-01321-8 (2023).

REVIEWER COMMENTS

Reviewer #1 (Remarks to the Author):

I have reviewed the authors' rebuttal. Overall I find their responses informative. I have a few remaining comments that concern how the responses are reflected in the revised ms:

In my review (#1) I made the following comment; "several crucial aspects of the report are not introduced. These include the use of the Cambridge Gambling Task to substantiate a differentiation between PC1 and PC2, and later on the use of ABCD and UKB cohorts in analyses of (epi)genetics and long-term outcomes".

The authors have added more methodological details in the SI, but I still think the readers will benefit from a brief (conceptual) elaboration in the Introduction of the principal logic of the 'bridge' from IMAGEN to ABCD and UKB, ideally noting some of the associated challenges.

Relatedly, the authors responded in some detail to my comments (#5-6) on the challenges related to bridging samples. They write:

"We agree that there exist strong assumptions when bridging IMAGEN to ABCD and UKB, as mentioned in our response to your previous question."

From what I can see, the authors have in the revision added some comments on the assumptions that go into the analyses. I would strongly encourage also adding some comments on these limitations to the Discussion section.

Only future within-person analyses will tell if the assumptions made are reasonable, but highlighting them in the Intro & Discussion will make the readers aware of this critical issue.

Reviewer #3 (Remarks to the Author):

This is a resubmission of a manuscript entitled: Structural neurodevelopment at the individual level - a life course investigation using ABCD, IMAGEN, and UK Biobank data by Shi

et al. Overall the premise of the manuscript is interesting including the attempt to characterize different data types across different timepoints across the lifespan. The authors have also responded fairly well to Reviewer comments. I have additional comments that are important to consider. They are listed in order of the paper as presented:

1. In the fourth line of the introduction the authors state that clinical symptoms of various disorders begin to emerge during adolescence which is true overall, but not true for all disorders. They list conduct disorder for one example, but symptoms of this disorder begin during childhood.

2. In the Results the authors state that PC1 was significantly associated with delay aversion, and list the p value as $p=0.30$. Presumably this is incorrect.

3. In the Results section the authors state that most items (spatial working memory, cambridge gambling etc) in comparing group 1 to group 3 with delayed neurodevelopment showing worse neurocognitive performance, specifically stating "but most of these items improved over time with brain maturation and became statistically equivalent at last follow up." What does statistically equivalent mean exactly - do they mean the groups were not significantly different in change over time ? The language needs to be precise regarding statistics.

4. In the second paragraph of this section the authors state "consistent with improvements of neurocognition, we observed decreased ADHD symptoms, but increased depression symptoms in Group 3". How exactly are increased depression symptoms consistent with improvements in neurocognition ? This does not make sense. Then in the same paragraph the authors state the opposite, that increased depression symptoms in Group 2 are consistent with worse neurocognitive performance over time. This starts to feel sloppy and in relation to almost justifying different patterns in the data.

5. In the fourth paragraph of this same section the authors state they "ask whether genetic variants could explain the delayed neurodevelopment and neurocognitive performances in this group". What do the authors mean when they talk about delayed neurodevelopment ?

What are the neurodevelopmental milestones measured and assessed ?

6. In the next section regarding genetic and epigenetic variation and structural neurodevelopment in the final few lines the authors talk about "nominal significance was observed for family affirmation". What is the threshold for "nominal significance". If the result is not significant after statistical correction, it should not be interpreted as significant and this section needs to be rewritten.

7. I worry about the UK Biobank interpretation of findings given that early signs of brain degeneration can start occurring in the decades of life examined using this data set. The brain changes can be dynamic in then opposite direction of neurodevelopment. There is no issue in examining certain genes in UK Biobank with brain structure, but the interpretation has to be taken in the context of what is happening the in the brain at that timepoint in the lifespan.

Reviewer #4 (Remarks to the Author):

This is the first time I have seen the manuscript. Bearing in mind existing reviews, I reserve most of my comments to authors' responses to the first round, and on the genetic and epigenetic aspects of the submitted work.

- The initial PCA methods are not sufficiently clear. Specifically why was 80% variance or 15 PCs deemed sufficient or optimal to be used in the k-means clustering? In response to Reviewer #1 point 5, the authors state that the rule of thumb to which they adhere here is the number of PCs to explain 75% variance. Presumably then, the 15th PC must explain $\geq 5\%$ of the slope variance – correct? If not, why include this PC if 14 (or perhaps fewer) PCs meet the stated 75% criterion. This should be tackled clearly in the manuscript.

- On a related note, a reviewer requested further information on the first 2 PCs, which I did not see was provided in the response. More generally, the PCA rotation matrix is provided, but the method of rotation was not given, nor were the standardised loadings / prop variance / cumulative variance on the rotated components (that is, we cannot see to what extent were the trajectories mainly explained by very many fewer components). Was this a

varimax rotation (sorry if I missed that)? Without this information, the nature of the PCs that are entered into the clustering cannot be clearly understood. This key information should be provided.

- When talking about the steps for 'translating' the IMAGEN-discovered groups into ABCD, for example, it would be informative to ensure that they are clearly labelled as longitudinal and cross-sectional datasets, as appropriate. Assumptions underlying the information about longitudinal trajectories (but not cross-sectional differences) during discovery being able to inform only cross-sectional differences in other samples are thorny and complex. Is it possible that the relationship between intercept and slope (and that meaningful differences in the ABCD intercepts) means the trajectories are only telling us what we could already glean from intercept differences?

- As the other reviewers have already pointed out, the authors take an interesting approach to understanding the genetic correlates of delayed brain development in the UKB sample. They now (response to Reviewer#2, point 5, for example) make clear what they are doing. It is very indirect: the polygenic score in UKL represents the genetic liability of having smaller cross-sectional ROI brain volumes - which are associated with differing trajectories of brain volumetric differences between 67 vs 1476 much younger participants. The authors should be crystal clear about what this represents and *how* it may be confounded. Validation of the GM patterns / groups in another longitudinal cohort would have been optimal.

- Please clarify whether cross-sectional or longitudinal FS pipelines were used across all (relevant) cohorts, and comment on the extent to which this is important for the reported results.

- Please provide further information on the analyses and the subsequent results that arise from the first part of the PRS (hitherto referred to as PGS – see minor comment below) analyses that corresponds to this part of the Methods: "optimal p-value thresholds were determined based on the best-fit R² using parent-rated psychiatric scores for ADHD and ASD, and the total WISCIV score. For EA, variants were selected using a P value threshold from 5e-08 to 1 with a step of 5e-05 and an average score under each P value threshold was

calculated.”

- Similarly, please provide supplementary results for the associations with group-re-weighted GMV by all thresholds of the PRS (PGS) tested – lines 577-579.

- Please include the results for the PRS for ADHD in Supplementary Table 9 alongside the other PRS results.

- I am confused about the EWAS approach (line 583-4). The prior analyses are all focussed on Group 1 vs 2 vs 3 analyses; and the authors explicitly put Groups 1 and 2 together because they share some similarities. As such, whereas it might be useful to use EWAS to describe the extent to which Groups 1 and 2 are similar epigenetically too, it seems odd to omit Group 1/2 vs Group 3 analyses entirely, given that the major thrust of the paper and conclusions concern Group 3 being of particular interest.

- Related to the above, the specific rationale for looking at Groups 1 and 2 separately for the EWAS component, having grouped them together previously, needs some further detail. Specify precisely what led you ‘...to reason that the differences of neurocognitive performances between Group 1 and Group 2 were quantitative (rather than qualitative), and might be subject to the effects of environmental exposure’. It would help to remind the reader of the specific quantitative evidence (e.g. effect sizes and adjust p-vals) that led you to this conclusion. Also, why couldn’t the differences with Group 3 also be subject to the effects of environmental exposure?

- More information should be provided on the statistical treatment of the adverse life events phenotype(s) in the Methods text. As far as I could see (and apologies if I had missed this) the only other indication about what had been done could be gleaned from Fig 4. Also, the authors should be commended for having undertaken mediation modelling in SEM – however, the point estimate and 95% CIs of the indirect effect (mediation effect, shown in Figure 4d) should be presented, rather than bars from 0.

- I am also concerned that the written articulation of the DNAm mediation results on

adverse environment \diamond brain associations overclaims on the basis of the statistical analyses. In the caption for Figure 4e, and in the conclusion of the paper (lines 307-309: “These results indicated that environmental exposure could contribute to disadvantages neurodevelopment and neurocognition by inducing epigenetic changes of neurogenesis-related genes”, see also lines 363-370), the mediation of cg06064461 is reported as being significant and/or meaningful. However, the authors undertook multiple mediation h-tests and it appears that the $p = 0.048$ was uncorrected, and does not survive multiple comparison testing. If the FDR-corrected results were non-significant, the authors should be cautious, clearly stating the small effect sizes and smaller proportion of that small effect that was non-significantly mediated by adverse environment. As it stands, I do not think that some of the abovementioned interpretations are supported by the results.

- With the complex analyses occurring across cross-sectional and longitudinal designs, it is important to be precise where possible; please make sure that terms like ‘changes’ are reserved for data that allows one to estimate change, and differences are used to refer to cross-sectional findings, throughout. E.g. line 249 : methylation was not measured longitudinally, so best to refer to differences or variation rather than changes. This also applies to the quote mentioned in the comment above (lines 307-309), where the correlational & cross-sectional data cannot directly support the assumption that the observed relationship between adverse environmental exposure and DNAm is causal).

- Minor – given the differing extent to which select groups wish to be considered to have ‘negative’ / less valued symptoms than those who are considered neurotypical, I suggest referring to polygenic scores (PGS) rather than polygenic *risk* scores (PRS) throughout. Either terms is widely used, but the latter suffers from being seen as possibly more negatively loaded.

Point-by-Point Response to the Reviewers' Comments

Runye Shi, Shitong Xiang, Tianye Jia, Trevor W. Robbins, ..., Gunter Schumann, Xiaolei Lin*, Barbara J. Sahakian*, Jianfeng Feng*, IMAGEN Consortium

Enclosed, please find the revised submission of the paper "Structural neurodevelopment at the individual level - a life-course investigation using ABCD, IMAGEN and UK Biobank data", for publication in Nature Communications. We are thankful to the comments from the reviewers and agree to all. Below we provide the point-by-point response to all reviewers.

Reviewer #1 (Remarks to the Author):

I have reviewed the authors' rebuttal. Overall, I find their responses informative. I have a few remaining comments that concern how the responses are reflected in the revised ms:

- 1) In my review (#1) I made the following comment; "several crucial aspects of the report are not introduced. These include the use of the Cambridge Gambling Task to substantiate a differentiation between PC1 and PC2, and later on the use of ABCD and UKB cohorts in analyses of (epi)genetics and long-term outcomes".

The authors have added more methodological details in the SI, but I still think the readers will benefit from a brief (conceptual) elaboration in the Introduction of the principal logic of the "bridge" from IMAGEN to ABCD and UKB, ideally noting some of the associated challenges.

Response: Thanks for the suggestion. We agree that including principal the logic of the bridge from IMAGEN to ABCD and UKB would benefit the readers in understanding the methods and results of the manuscript. We have now added a brief clarification regarding the bridge from IMAGEN to ABCD and UKB in lines 141-151 under the Introduction section:

"It is worth noting that, in order to extend the investigation from adolescence to late childhood and mid-to-late adulthood, we bridged IMAGEN to Adolescent Brain Cognitive Development study (ABCD) and UK Biobank (UKB) through different mapping approaches assuming population homogeneity. Specifically, longitudinal brain changes were mapped to baseline neuroimaging phenotypes in IMAGEN, which were further used to evaluate the associations between cross-sectional brain measures and population cluster in ABCD, assuming comparable linear changes from late childhood to adolescence for each structural brain measure. Genomic and neuroimaging data in ABCD allowed us to identify potential genetic variations associated with particular population clusters. Finally, genomic, neuroimaging and other related phenotypes in UKB allowed us to investigate the long-term impact of genetic-proxied neurodevelopment."

- 2) Relatedly, the authors responded in some detail to my comments (#5-6) on the challenges related to bridging samples. They write:

"We agree that there exist strong assumptions when bridging IMAGEN to ABCD and UKB, as mentioned in our response to your previous question."

From what I can see, the authors have in the revision added some comments on the assumptions that go into the analyses. I would strongly encourage also adding some comments on these limitations to the Discussion section.

Only future within-person analyses will tell if the assumptions made are reasonable, but highlighting them in the Intro & Discussion will make the readers aware of this critical issue.

Response: Thanks for the suggestion. We have now added the comments on assumptions / limitations of current approach and highlighted the importance of validation using future within-person analyses once long-term follow-up data of these adolescents become available in lines 401-415 of the Discussion section:

*"Although we tried to link the neurodevelopmental patterns from IMAGEN to ABCD and UKB, this mapping using genetic and neuroimaging associations may subject to confounding bias.....In addition, both the appropriateness of using the proxy phenotype and results of GWAS conducted in ABCD were successfully validated. **However, the robustness of the bridge approach used in this study and its assumptions still await further validation once follow-up data become available for the ABCD participants. Meanwhile, long-term follow-ups of the socio-economic outcomes in IMAGEN adolescents are needed to validate our results obtained from UK Biobank. In other words, large-scale longitudinal data that span the entire life-course may confirm the reliability of the findings obtained in our study.**"*

Reviewer #3 (Remarks to the Author):

This is a resubmission of a manuscript entitled: Structural neurodevelopment at the individual level - a life course investigation using ABCD, IMAGEN, and UK Biobank data by Shi et al. Overall the premise of the manuscript is interesting including the attempt to characterize different data types across different timepoints across the lifespan. The authors have also responded fairly well to Reviewer comments. I have additional comments that are important to consider. They are listed in order of the paper as presented:

- 3) In the fourth line of the introduction the authors state that clinical symptoms of various disorders begin to emerge during adolescence which is true overall, but not true for all disorders. They list conduct disorder for one example, but symptoms of this disorder begin during childhood.

Response: Thanks for the comment and we apologize for the confusion. Due to varying definitions of childhood and adolescence, it is often difficult to distinguish between late childhood and early adolescence. For example, as suggested in Blakemore¹, substance use disorder (sub-category of conduct disorders) often starts to emerge in adolescence. To avoid further confusion, we have revised this sentence to better reflect the increasing risk of these disorders during adolescent brain development:

"The risk for many neuropsychiatric disorders increases during this period, including conduct disorder, mood disorder and schizophrenia"

- 4) In the Results the authors state that PC1 was significantly associated with delay aversion, and list the p value as p=0.30. Presumably this is incorrect.

Response: Thanks for the comment. We apologize for the typo and have corrected the text with p value 0.030 in the revised manuscript lines 168-169.

"...where PC1 was significantly associated with delay aversion ($r = 0.07$, $P_{adj} = 0.030$) and risk adjustment ($r = -0.08$, $P_{adj} = 0.020$)..."

- 5) In the Results section the authors state that most items (spatial working memory, cambridge gambling etc) in comparing group 1 to group 3 with delayed neurodevelopment showing worse neurocognitive performance, specifically stating "but most of these items improved over time with brain maturation and became statistically equivalent at last follow up." What does statistically equivalent mean exactly - do they mean the groups were not significantly different in change over time? The language needs to be precise regarding statistics.

Response: Thanks for the question. By stating that these items became statistically equivalent at the last follow up, we meant that the two-tailed t-test for these neurocognitive measurements at the last follow-up (comparing group 3 vs group 1) were not significant after FDR correction. In statistical language, let μ_1 and μ_2 represent the mean neurocognitive score of each item for adolescents in

group 1 and group 3, respectively. The hypothesis testing against H_0 , where $H_0: \mu_1 = \mu_2$ and $H_1: \mu_1 \neq \mu_2$, achieved p value larger than 0.05 after FDR correction, and thus the null hypothesis H_0 could not be rejected. It actually compared the cross-sectional performance of groups 1 and 3 at the last follow up, rather than longitudinal comparisons. We apologized for potential misunderstanding and have now revised the manuscript in lines 216-217 of the Results section.

*"...but most of these items improved over time with brain maturation and became statistically equivalent (**two-tailed t-test: $P_{adj} > 0.05$**) at the last follow-up..."*

- 6) In the second paragraph of this section the authors state "consistent with improvements of neurocognition, we observed decreased ADHD symptoms, but increased depression symptoms in Group 3". How exactly are increased depression symptoms consistent with improvements in neurocognition? This does not make sense. Then in the same paragraph the authors state the opposite, that increased depression symptoms in Group 2 are consistent with worse neurocognitive performance over time. This starts to feel sloppy and in relation to almost justifying different patterns in the data.

Response: Thanks for the question. We apologize for the confusing wording that led to potential misunderstanding. The sentence "but increased depression symptoms in Group 3" actually referred to the inconsistency between the improvement of neurocognition and increased depression symptoms in Group 3. We apologize again for the inappropriate use of 'but'. We have revised this expression in lines 217 of the Results section:

*"Consistent with the improvements of neurocognition, we observed decreased attention-deficit/hyperactivity disorder (ADHD) symptom. **However, in contrast to improved neurocognition, we observed increased depression symptoms in Group 3** (Fig. 2b and Supplementary Table 4)."*

- 7) In the fourth paragraph of this same section the authors state they "ask whether genetic variants could explain the delayed neurodevelopment and neurocognitive performances in this group". What do the authors mean when they talk about delayed neurodevelopment? What are the neurodevelopmental milestones measured and assessed?

Response: Thanks for the question. Clinically, neurodevelopment delay often referred to delayed development of skills in infants and young children, which was not recorded in the IMAGEN cohort. When consider the development of brain morphology, a delayed neurodevelopment pattern refers to a slower pace at which developmental milestones are attained. According to Bethlehem et al.², total gray matter volume (GMV) increases from mid-gestation onwards, peaking at childhood, and nonlinearly decreases throughout adolescence. Therefore, the pace of structural neurodevelopment could be assessed using the time of peak GMV. As showed in Supplementary Fig. 6, participants in Group 3 reached the peak of total GMV at a later time than those in Groups 1 and 2.

Supplementary Fig. 6. Estimated total GMV developmental curves of groups from 5y to 37y. Mean total GMV developmental trajectories (with 95% confidence bands) for groups were plotted using estimated individual GMV trajectories in Supplementary Fig. 5. Ranges from the 2.5th percentile to the 97.5th percentile of the corresponding group were plotted as bands.

Thus, we concluded that participants in Group 3 showed delayed neurodevelopment. To better elucidate delayed neurodevelopment, we have included a brief clarification where it was first introduced in lines 191-195 of the Results section:

*"Consistently we observed continuously decreasing GMV in Group 1 and Group 2 (with slower rate of GMV decrease in Group 2), and increasing GMV in Group 3 for most ROIs (Fig. 1c), indicating delayed neurodevelopment and brain maturation in Group 3 compared to the other groups, **where delayed neurodevelopment was proxied using later peaking time of total GMV.**"*

- 8) In the next section regarding genetic and epigenetic variation and structural neurodevelopment in the final few lines the authors talk about "nominal significance was observed for family affirmation". What is the threshold for "nominal significance". If the result is not significant after statistical correction, it should not be interpreted as significant and this section needs to be rewritten.

Response: Thanks for the comment. Nominal significance referred to smaller unadjusted p values than 0.05. Considering the low-to-medium correlations among multiple environmental exposures (Supplementary Fig. R1), the Bonferroni approach to correct for multiple tests could lead to conservative p values and low statistical power. The fact that a single methylation site cg06064461 was identified to mediate the effect of family affirmation on neurodevelopment at nominal significance level provided evidence toward promising future research directions. Therefore, we included this result in the main manuscript with the hope that it could benefit future research. However, we totally agree that more cautious language should be used here regarding the statistical significance. We have revised the manuscript accordingly in lines 314-421 of the Results section:

"Overall, no mediation effects of cg06064461 methylation on the environment - neurodevelopment pathway showed statistical significance after correcting for

multiple testing (Fig. 4d and Supplementary Table 12). However, given that only one site could be identified with differential methylation between Groups 1 and 2, it should be noted that an uncorrected significance was observed for family affirmation, where higher levels of family affirmation were associated with higher peak GMV through reduced cg06064461 methylation ($\beta = 0.005$, mediation proportion = 0.09, $P_{unadj} = 0.048$, $P_{adj} = 0.191$) (Fig. 4e)."

Supplementary Fig. R1. Correlation matrix among all environmental exposures.

- 9) I worry about the UK Biobank interpretation of findings given that early signs of brain degeneration can start occurring in the decades of life examined using this data set. The brain changes can be dynamic in then opposite direction of neurodevelopment. There is no issue in examining certain genes in UK Biobank with brain structure, but the interpretation has to be taken in the context of what is happening the in the brain at that timepoint in the lifespan.

Response: Thanks for your suggestion. We totally agree that it is exceedingly challenging to entangle the potential relationships between neurodevelopment and neurodegeneration due to their complex manifestations. As a result, it is also difficulty to interpret the relationship between genetically-predicted delayed neurodevelopment and brain structural morphology in mid-to-late adulthood using data from UK Biobank. We didn't talk much about this finding in the original manuscript, but it could be informative to add some of these comments. Thus, we have included a brief explanation in lines 348-352 of the Results section:

"Findings of a negative correlation between PGS and lower GMV in these regions could be interpreted as either continued influence of delayed neurodevelopment, effects from genetically-related environmental exposures or genetically-related neurodegenerative processes. Further studies are needed to explore and disentangle the potential underlying biological mechanisms."

In addition, we also highlighted the importance of further longitudinal exploration for the effects of delayed adolescent neurodevelopment on long-term outcomes and mid-to-late adulthood brains

measures in lines 412-415 of the Discussion section:

"Meanwhile, long-term follow-ups of the socio-economic outcomes in IMAGEN adolescents are needed to validate our results obtained from UK Biobank. In other words, large-scale longitudinal data that span the entire life-course may confirm the reliability of the findings obtained in our study."

Reviewer #4 (Remarks to the Author):

This is the first time I have seen the manuscript. Bearing in mind existing reviews, I reserve most of my comments to authors' responses to the first round, and on the genetic and epigenetic aspects of the submitted work.

- 10) The initial PCA methods are not sufficiently clear. Specifically, why was 80% variance or 15 PCs deemed sufficient or optimal to be used in the k-means clustering? In response to Reviewer #1 point 5, the authors state that the rule of thumb to which they adhere here is the number of PCs to explain 75% variance. Presumably then, the 15th PC must explain $\geq 5\%$ of the slope variance – correct? If not, why include this PC if 14 (or perhaps fewer) PCs meet the stated 75% criterion. This should be tackled clearly in the manuscript.

Response: Thank you for the comment and we apologize for any confusion due to lack of method description concerning the selection of PCs. We now include the standard deviation, standardized loadings, proportion of variance explained and cumulative proportion of variance explained for the first 15 PCs used for multivariate clustering in the Supplementary Table 1. As shown in this table, only the first 2 PCs explained more than 5% of total slope variation and the minimum number of PCs explaining more than 75% cumulative variance was 13. The goal of PCA was to reduce dimensions and yet maintain robust results as compared to using the original neuroimaging data. Supplementary Table R1 showed the clustering results when using 13, 14, 15, 16, and 17 PCs, where optimal number of clusters remained stable when we select 15 or more PCs, suggesting that the remaining PCs after the 15th PC offered limited information about the original neuroimaging data. Therefore, considering both dimension reduction and result robustness, 15 PCs were selected for the multivariate clustering. To clarify the approach on PC selection, we revised the Methods section as follows:

"Considering both the proportion of cumulative variance explained and robustness of multivariate clustering results, The first 15 principal components (Supplementary Table 1), which explained 80% of the total variation, were used in the multivariate k-means clustering. The optimal number of clusters was selected based on the Elbow method with the constraint that each cluster contain at least 4% of the overall population."

Added information in Supplementary Table 1. Rotation matrix and proportion of variance explained by each PCA component.

	PC1	PC2	PC3	PC4	PC5	PC6	PC7	PC8	PC9	PC1	PC1	PC1	PC1	PC1	PC1
										0	1	2	3	4	5
Cortical regions															
bankssts	0.04	0.04	-0.03	0.02	-0.04	0.16	-0.09	0.12	-0.19	0.09	-0.16	0.10	0.00	-0.09	-0.09
caudalanteriorcingulate	0.04	0.02	-0.02	-0.09	0.03	0.05	0.25	-0.19	0.17	0.08	-0.23	-0.10	0.12	0.20	-0.02
caudalmiddlefrontal	0.04	-0.10	-0.01	-0.14	0.03	0.01	0.13	0.04	0.09	0.08	0.13	-0.07	-0.07	-0.08	-0.01
cuneus	0.03	-0.11	0.09	0.19	0.21	-0.16	-0.17	-0.10	0.06	-0.12	-0.04	-0.09	0.00	-0.04	0.31
entorhinal	0.03	0.08	-0.11	0.05	0.08	0.15	-0.04	-0.09	0.28	-0.23	0.22	0.31	-0.10	-0.01	-0.11
fusiform	0.04	0.12	-0.08	0.07	0.08	0.05	-0.13	0.06	-0.04	-0.03	-0.09	0.03	0.09	-0.05	0.02

Cumulative proportion	0.41	0.47	0.52	0.56	0.59	0.62	0.65	0.67	0.69	0.71	0.73	0.75	0.76	0.78	0.79
------	------	------	------	------	------	------	------	------	------	------	------	------	------	------

* Only standardized loadings were presented here while both loadings and standardized loadings were included in the Supplementary Table 1.

Supplementary Table R1. The clustering results (# of cluster = 3) using different numbers of PCs

Using 15 PCs	Using 13 PCs			Using 14 PCs		
	Group 1 (n=710)	Group 2 (n=766)	Group 3 (n=67)	Group 1 (n=706)	Group 2 (n=770)	Group 3 (n=67)
Group 1 (n=711)	708	3	0	706	5	0
Group 2 (n=765)	2	763	0	0	765	0
Group 3 (n=67)	0	0	67	0	0	67

11) On a related note, a reviewer requested further information on the first 2 PCs, which I did not see was provided in the response. More generally, the PCA rotation matrix is provided, but the method of rotation was not given, nor were the standardised loadings / prop variance / cumulative variance on the rotated components (that is, we cannot see to what extent were the trajectories mainly explained by very many fewer components). Was this a varimax rotation (sorry if I missed that)? Without this information, the nature of the PCs that are entered into the clustering cannot be clearly understood. This key information should be provided.

Response: Thank you for your question and suggestion. We apologize for the potential misinterpretation of the question raised by reviewer #1. The first 2 PCs explained 40.83% and 6.09% of the total slope variation and we agree that the rotation matrix should be included for better understanding of these PCs. We have now included the standard deviation, standardized loadings, proportion of variance explained and cumulative proportion of variance explained by each of the first 15 PCs in the Supplementary Table 1. In brief, PC1 and PC2 both represent combinations of GMV trajectories over the entire brain that were associated with baseline total GMV, but with different associations with items in CGT. From ad-hoc analyses, we observed association between PC1 and delay aversion / risk adjustment, and between PC2 and deliberation time / overall betting / risk taking.

Principal components were derived by carrying out singular value decomposition of centered GMV trajectories estimated from the neuroimaging data (a varimax rotation method), where the rotation matrix was obtained from the right singular vector. We have modified the Methods section with a more detailed description of the PCA.

"Dimension reduction via PCA (prcomp function in the stats 4.2.2 package) was performed on standardized individual GMV trajectories estimated using neuroimaging data of 44 ROIs. The rotation matrix was obtained from the right singular vector, where singular value decomposition was performed on the centered GMV trajectories."

- 12) When talking about the steps for ‘translating’ the IMAGEN-discovered groups into ABCD, for example, it would be informative to ensure that they are clearly labelled as longitudinal and cross-sectional datasets, as appropriate. Assumptions underlying the information about longitudinal trajectories (but not cross-sectional differences) during discovery being able to inform only cross-sectional differences in other samples are thorny and complex. Is it possible that the relationship between intercept and slope (and that meaningful differences in the ABCD intercepts) means the trajectories are only telling us what we could already glean from intercept differences?

Response: Thank you for the comment. To remind the readers of the type of dataset used in each cohort, we have modified lines 267-270 of the results section:

*"Specifically, we began by calculating the ROI-specific weight in discriminating Group 3 (relative to Groups 1/2) in IMAGEN **using baseline neuroimaging data** adjusting for potential confounders, and applying these weights to corresponding ROIs in ABCD **baseline data** to obtain the Group3-reweighted GMV, which was then used as the proxy phenotype in the Group 3 GWAS (Methods)."*

Your question regarding the mapping from longitudinal GMV trajectory to cross-sectional GMV data was crucial since the other reviewers also raised the same question during the previous review correspondence. We have briefly talked about the mapping approaches and its related limitations in the Discussion section in the revised manuscript in answering reviewer # 1.

We agree that, although a strong correlation was observed between baseline GMV and longitudinal GMV trajectory in IMAGEN ($r = -0.68$, $P < 0.001$, Supplementary Fig. 1), longitudinal neuroimaging data were desired in predicting the group labels in ABCD study. As shown in Supplementary Fig. 1, participants with the same level of baseline total GMV could have opposite growth direction in follow-ups due to individual heterogeneity that cannot be explained by observed covariates. However, the follow-up neuroimaging data in ABCD study were not currently available and there is a lack of other large-scale neuroimaging cohorts that can be used to investigate the genetic variation between group 3 and groups 1/2. Therefore, the approach we used to address this, was to construct a prediction model that mapped the longitudinal GMV trajectory to baseline GMV in IMAGEN, such that this model was able to predict the group level using baseline data in ABCD. Given the excellent performance of the prediction model (AUC = 0.98 for discriminating Group 3 vs Group 1/2), we believe that this is the best one can do considering that longitudinal follow-up data were not available for ABCD. Nevertheless, we acknowledge that the robustness of this mapping approach needs validation once long-term follow-up data for the ABCD study become available.

Supplementary Fig. 1. The joint distribution of baseline total GMV and GMV developmental trajectories.

- 13) As the other reviewers have already pointed out, the authors take an interesting approach to understanding the genetic correlates of delayed brain development in the UKB sample. They now (response to Reviewer#2, point 5, for example) make clear what they are doing. It is very indirect: the polygenic score in UKL represents the genetic liability of having smaller cross-sectional ROI brain volumes - which are associated with differing trajectories of brain volumetric differences between 67 vs 1476 much younger participants. The authors should be crystal clear about what this represents and *how* it may be confounded. Validation of the GM patterns / groups in another longitudinal cohort would have been optimal.

Response: Thank you for the comment and suggestion. We agree that there should be a cautious approach to the interpretation of the negative correlations between the genetic liability of delayed neurodevelopment and cross-sectional brain volumes in mid-to-late adulthood, since cross-sectional brain measures among participants in UKB contain a mixture of information regarding the long-term influence of neurodevelopment and the process of neurodegeneration. We have now included a brief discussion regarding this finding in lines 348-352 of the Results section:

" Findings of a negative correlation between PGS and lower GMV in these regions could be interpreted as either continued influence of delayed neurodevelopment, effects from genetically-related environmental exposures or genetically-related neurodegenerative processes. Further studies are needed to explore and disentangle the potential underlying biological mechanisms."

In addition, we agree that validation of our findings using another longitudinal adolescent cohort is required in the future when such a dataset should eventually become available through the ABCD study. However, due to the lack of large-scale longitudinal neuroimaging cohort, validation of the current findings was not possible. We have highlighted the importance of validation cohorts and the limitations of the current study in the Discussion section:

"However, the robustness of the bridge approach used in this study and its assumptions still await further validation once follow-up data become available for the ABCD participants. Meanwhile, long-term follow-ups of the socio-economic

outcomes in IMAGEN adolescents are needed to validate our results obtained from UK Biobank. In other words, large-scale longitudinal data that span the entire life-course may confirm the reliability of the findings obtained in our study."

- 14) Please clarify whether cross-sectional or longitudinal FS pipelines were used across all (relevant) cohorts, and comment on the extent to which this is important for the reported results.

Response: Thank you for your suggestion. We have included the clarification that cross-sectional FS pipelines were used in all cohorts under both the Methods section and **Supplementary Methods** for ABCD, IMAGEN, HCP and PNC (except UKB since the imaging-derived phenotypes in UKB were provided by the UKB study group).

Methods section:

"Assessment of regional morphometric structure were extracted by FreeSurfer v6.0 **cross-sectional pipelines** using Desikan-Killiany (h.aparc) atlas for cortical regions, and ASEG atlas for subcortical regions."

In addition, we included the reason why we chose to use cross-sectional FS pipelines in the preprocessing of the IMAGEN structural neuroimaging data in the **Supplementary Methods**, since only IMAGEN involved longitudinal dataset in our study.

"For consistency, cross-sectional rather than longitudinal FreeSurfer pipelines were used, including in the longitudinal dataset, as not every individual was measured on all three follow-ups. Therefore, data were preprocessed using cross-sectional FS pipelines and post-hoc analyses taking into account the within-individual correlations were conducted using mixed effect models"

- 15) Please provide further information on the analyses and the subsequent results that arise from the first part of the PRS (hitherto referred to as PGS – see minor comment below) analyses that corresponds to this part of the Methods: “optimal p-value thresholds were determined based on the best-fit R2 using parent-rated psychiatric scores for ADHD and ASD, and the total WISCIV score. For EA, variants were selected using a P value threshold from 5e-08 to 1 with a step of 5e-05 and an average score under each P value threshold was calculated.”

Response: Thank you for the suggestion. We have included more detailed information illustrating how polygenic risk scores were calculated and compared in the Methods section and Supplementary Fig. 18.

"...For ADHD, ASD and IQ, optimal p-value thresholds were determined based on the best-fit R2 using parent-rated psychiatric scores for ADHD and ASD, and the total WISCIV score (Supplementary Fig. 17). For EA, variants were selected using a P value threshold from 5e-08 to 1 with a step of 5e-05 and an average score under each P value threshold was calculated. **One-way ANOVA test with Fisher's Least Significant Difference (LSD) post-hoc test was used to compare PGS among groups."**

Supplementary Fig. 18. Polygenic risk scores (PGS) were used to predict genetic liability to ADHD, ASD and IQ in IMAGEN. Total variance in corresponding traits in IMAGEN explained by the PGS for multiple p value thresholds was shown. The red bar indicated the optimal p value threshold explaining the maximum amount of variance.

16) Similarly, please provide supplementary results for the associations with group-re-weighted GMV by all thresholds of the PRS (PGS) tested – lines 577-579.

Response: Thank you for the suggestion. We have now included Supplementary Fig. 22 to show PGS tested with group-re-weighted GMV.

Supplementary Fig. 22. Polygenic risk scores (PGS) were used to distinguish Group 3 from Group 1/2 (a) and Group 2 from Group 1 (b) in IMAGEN. Total variance in the Group3-reweighted GMV and Group2-reweighted GMV in IMAGEN explained by the PGS for multiple p value thresholds was shown. The red bar indicated the optimal p value threshold explaining the maximum amount of variance.

17) Please include the results for the PRS for ADHD in Supplementary Table 9 alongside the other PRS results.

Response: Thank you for the suggestion. We have now included the comparison of PGS for ADHD among groups in Supplementary Table 9. We also found a typographical error in the main manuscript and have corrected it in lines 252-253:

"Group 3 had higher PGS for ADHD than both Group 1 ($P_{adj} = 0.007$) and Group 2 ($P_{adj} = 0.017$), while Group 2 was not statistically different from Group 1 ($P_{adj} = 0.42 \rightarrow 0.424$)."

Supplementary Table 9. Comparison of PGS for attention-deficit/hyperactivity disorder (ADHD), autism spectrum disorder (ASD), educational attainment (EA) and IQ among three groups.

Term	Group 2 vs Group 1		Group 3 vs Group 1		Group 3 vs Group 2	
	t	P_{adj}	t	P_{adj}	t	P_{adj}
ADHD	-0.799	0.424	2.709	0.007	2.392	0.017
ASD	-0.359	0.720	0.476	0.634	0.623	0.533
EA	-0.426	0.670	1.105	0.270	1.281	0.200
IQ	1.579	0.115	0.656	0.512	0.015	0.988

18) I am confused about the EWAS approach (line 583-4). The prior analyses are all focused on Group 1 vs 2 vs 3 analyses; and the authors explicitly put Groups 1 and 2 together because they share some similarities. As such, whereas it might be useful to use EWAS to describe the

extent to which Groups 1 and 2 are similar epigenetically too, it seems odd to omit Group 1/2 vs Group 3 analyses entirely, given that the major thrust of the paper and conclusions concern Group 3 being of particular interest.

Response: Thank you for the comments. We have previously performed an EWAS between Group 3 and Groups 1/2. However, no genome-wide significant methylation site could be identified. Moreover, a preliminary analysis considering baseline socioeconomic and family environment showed that, more environmental differences were observed between group 1 and group 2, rather than between group 3 and groups 1/2. Therefore, we emphasized the impact of environmental factors in explaining the differences between group 1 and group 2. While the identification of group 3 is interesting in terms of genetic variation, the findings regarding the differences between group 1 and 2 epigenetically were also crucial. We have included the EWAS results between Group 3 and Groups 1/2 in the Supplementary Fig. 14, and modified lines 326-327 of the Results section,

" Furthermore, no significant site was identified in the EWAS investigating Group 3 versus Groups 1/2 (Supplementary Fig. 14)."

and lines 611-620 of the Methods section.

"EWAS was performed among Group 1 (n = 446), Group 2 (n = 463) and Group 3 (n = 36) in IMAGEN. Methylation data were collected using the Illumina Infinium HumanMethylation450 BeadChip. Locus-specific genome-wide methylation analysis was conducted and beta values at each Autosomal CpG site were used in pairwise comparisons with group label as the phenotype using logistic regression adjusting for sex, experimental batches (recruitment center and acquisition wave), the first two principal components of methylation composition and the first four principal components of estimated differential cell counts. We used Synthetic Minority Oversampling Technique (SMOTE) (smote function in performanceEstimation 1.1.0 package; default setting) to address the issue of class imbalance when comparing Group 3 with others."

Supplementary Fig. 14. EWAS Manhattan plot in the IMAGEN population comparing Group 3 with Groups 1/2. Group 1 (n=446) and Group 2 (n=463) (relative to Group 3, n=36) status was used as the phenotype, adjusting for potential confounders. No significant site was identified in the EWAS investigating Group 3 versus Groups 1/2.

19) Related to the above, the specific rationale for looking at Groups 1 and 2 separately for the EWAS component, having grouped them together previously, needs some further detail. Specify precisely what led you ‘...to reason that the differences of neurocognitive performances between Group 1 and Group 2 were quantitative (rather than qualitative), and might be subject to the effects of environmental exposure’. It would help to remind the reader of the specific quantitative evidence (e.g. effect sizes and adjust p-val) that led you to this conclusion. Also, why couldn’t the differences with Group 3 also be subject to the effects of environmental exposure?

Response: Thank you for your suggestions. We agree that providing the rationale regarding the combination of groups 1/2 would benefit readers in understanding the analytical results better. In summary, adolescents in group 3 exhibited differences in both magnitude and timing of GMV peak compared to those in groups 1/2, while those in group 2 exhibited differences only in the magnitude of peak GMV compared to group 1. In addition, adolescents in groups 3 exhibited increasing GMV growth during 14y - 23y, while those in groups 1/2 had decreasing GMV during this period, with magnitude higher in group 1 than in group 2. All these results indicated an important pattern of

delayed neurodevelopment in group 3, as well as quantitative differences between group 1 and 2. Due to similarities when comparing group 3 vs groups 1/2 (Supplementary Fig. 19), we combined adolescents in group 1 and 2 to increase statistical power. This also helped to better formulate the research question concerning genetic variation associated with delayed neurodevelopment.

We aimed to understand what contributed to the differences between group 1 and 2, as well as between group 3 and groups 1/2. Group 2 GWAS did not identify significant genetic signals, and, in addition, homogeneous genetic liability for neurodevelopmental disorders and related traits (ADHD, ASD and IQ) were observed for adolescents in groups 1/2. Therefore, environmental exposure is likely to contribute to the quantitative differences between group 1 and 2, especially, given the evidence of epigenetic findings and socioeconomic differences at baseline (e.g. socioeconomic/housing, health and relationship/addiction scores and family affirmation in the Family Stress Scale and Family Life Questionnaire). To illustrate this reasoning process more explicitly, we modified lines 294-302 in the Results section:

"No genome-wide significant SNPs were identified in the Group 2 GWAS (Supplementary Fig. 13). However, the large overlap between the neurodevelopmental patterns and homogeneous genetic liability for neurodevelopmental disorders and related traits (ADHD, ASD, IQ and EA) in Groups 1/2 led us to reason that the differences of neurocognitive performances between Group 1 and 2 were quantitative (rather than qualitative) and might be due to the effects of environmental exposure. This was also supported by the baseline differences in socioeconomic and family factors, such as stressor scores of socioeconomic/housing ($d = 0.30$, $P_{adj} < 0.001$), health ($d = 0.16$, $P_{adj} = 0.014$), relationship/addiction ($d = 0.29$, $P_{adj} < 0.001$) and family affirmation ($d = -0.11$, $P_{adj} = 0.045$) in Group 2 versus Group 1."

The findings that no genome-wide significant methylation site was identified in the EWAS comparing Group 3 versus groups 1/2 (Supplementary Fig. 14) could be explained by either the fact that there are no epigenetic differences between them, or that the effect size was too small to be detected due to the small sample size. To better elucidate this interpretation, we provided some comments in lines 391-395 under the Discussion section.

"However, it does not necessarily mean that the differences between Group 3 and Group 1 could only be attributed to genetic variation, or that differences between Group 2 and Group 1 was purely due to environment. Future research with larger sample sizes and adequate statistical power are needed to elucidate the potential interplay between gene and environment on structural brain development."

Supplementary Fig. 19. Miami plot of GWAS for Group 3 vs Group 1 and Group 2.

- 20) More information should be provided on the statistical treatment of the adverse life events phenotype(s) in the Methods text. As far as I could see (and apologies if I had missed this) the only other indication about what had been done could be gleaned from Fig 4. Also, the authors should be commended for having undertaken mediation modelling in SEM – however, the point estimate and 95% CIs of the indirect effect (mediation effect, shown in Figure 4d) should be presented, rather than bars from 0.

Response: Thank you for your comments. All the calculations of negative/positive life event scores were provided in the **Supplementary Methods**. We have added all the environmental exposures used in the mediation analyses in lines 621-633 in the Methods section.

*"Next, we aimed to investigate the association between CpG site and gene methylation with environmental factors of interest. We conducted mediation analyses (sem function in the lavaan 0.6-12 package) and estimated the total effect of **childhood environmental exposures** on estimated peak GMV and the indirect effect mediated by cg06064461 hypermethylation. Sex, batches effects, methylation composition components and differential cell count components were included as covariates. Total, direct and indirect effects and their standard deviations were estimated using 1000-iterated nonparametric bootstrap approach. False discovery rate (FDR) was used to correct for multiple testing within scales. **Childhood environmental exposures included abuse (physical/emotional/sexual) and neglect (physical/emotional) scores in the CTQ, socioeconomics/housing, work/pressure, health and relationship/addiction scores in DAWBA-Family Stress Scale, and affirmation, discipline, rules and special allowance scores in the DAWBA-Family Life Questionnaire. Details about the calculation of each environmental exposure score are presented in the***

Supplementary Methods.”

As suggested, we have added both the point estimate and 95% CIs of the indirect effects in Fig. 4(d).

Fig. 4(d) Proportion of the mediation effects through cg06064461 methylation in the environmental exposure - peak GMV pathway, adjusting for potential confounders. **Error bars indicate 95% confidence intervals of the estimated mediation proportion.** Environmental factors were sorted by P values of the mediation effects. No mediation effects of cg06064461 methylation showed statistical significance after correcting for multiple testing, although uncorrected significance was observed between family affirmation and peak GMV. Childexp, child’s experience of family life; FamStress, family stressors; CTQ, Childhood Trauma Questionnaire.

21) I am also concerned that the written articulation of the DNAm mediation results on adverse environment - brain associations overclaims on the basis of the statistical analyses. In the caption for Figure 4e, and in the conclusion of the paper (lines 307-309: “These results indicated that environmental exposure could contribute to disadvantages neurodevelopment and neurocognition by inducing epigenetic changes of neurogenesis-related genes”, see also lines 363-370), the mediation of cg06064461 is reported as being significant and/or meaningful. However, the authors undertook multiple mediation h-tests and it appears that the $p = 0.048$ was uncorrected, and does not survive multiple comparison testing. If the FDR-corrected results were non-significant, the authors should be cautious, clearly stating the small effect sizes and smaller proportion of that small effect that was non-significantly mediated by adverse environment. As it stands, I do not think that some of the abovementioned interpretations are supported by the results.

Response: Thank you for the suggestions. Firstly, considering the low-to-medium correlations between environmental exposures, the Bonferroni correction could lead to conservative conclusions with reduced statistical power. Secondly, environment factors usually impact related phenotypes via DNA methylation at multiple gene regions, and a single methylation site could also be influenced

by multiple environmental exposures. Given the results that a single methylation site cg06064461 was identified to mediate the effect of family affirmation on neurodevelopment at an uncorrected significance level with relatively a small sample size (n = 446 for group 1, n = 463 for group 2). This may provide strong evidence toward promising future research directions. However, we totally agree that more cautious language should be used here regarding the statistical significance. Therefore, we have revised the manuscript accordingly in the caption of Figure 4e and in lines 314-421 of the Results section:

*"Fig. 4(e) Mediation model was conducted to analyze the direct and indirect effect of family affirmation on peak GMV, with cg06064461 methylation as the mediator. Results showed that cg06064461 methylation significantly mediate the relationship between family affirmation and peak GMV ($\beta = 0.005$, mediation proportion = 9.26%, * $P_{unadj} = 0.048$, $P_{adj} = 0.191$)."*

"Overall, no mediation effects of cg06064461 methylation on the environment - neurodevelopment pathway showed statistical significance after correcting for multiple testing (Fig. 4d and Supplementary Table 12). However, given that only one site could be identified with differential methylation between Groups 1 and 2, it should be noted that an uncorrected significance was observed for family affirmation, where higher levels of family affirmation were associated with higher peak GMV through reduced cg06064461 methylation ($\beta = 0.005$, mediation proportion = 0.09, $P_{unadj} = 0.048$, $P_{adj} = 0.191$) (Fig. 4e)."

And we also modified lines 323-326 with more cautious language in the interpretation of this finding:

"These results indicated that environmental exposure could contribute to disadvantaged neurodevelopment and neurocognition by inducing epigenetic changes of neurogenesis-related genes. However, only a small proportion of the mediation effect was identified given the relatively small sample size."

- 22) With the complex analyses occurring across cross-sectional and longitudinal designs, it is important to be precise where possible; please make sure that terms like 'changes' are reserved for data that allows one to estimate change, and differences are used to refer to cross-sectional findings, throughout. E.g. line 249 : methylation was not measured longitudinally, so best to refer to differences or variation rather than changes. This also applies to the quote mentioned in the comment above (lines 307-309), where the correlational & cross-sectional data cannot directly support the assumption that the observed relationship between adverse environmental exposure and DNAm is causal).

Response: Thank you for your suggestion. We have checked all the related term as required and revised the inappropriate term accordingly.

lines 95-97

"Compared to Group 1, Group 2 exhibited a slower rate of GMV decrease and worsened

*neurocognitive development, which was associated with epigenetic **differences** and greater environmental burden."*

lines 140-142

*"Both genome-wide and epigenome-wide association studies are conducted to dissect the genetic and epigenetic **variations** associated with each cluster."*

line 259

*"**Genetic and epigenetic variations** contribute to structural neurodevelopment."*

lines 323-325

*"These results indicated that environmental exposure could contribute to disadvantaged neurodevelopment and neurocognition by inducing epigenetic **differences** of neurogenesis-related genes."*

lines 383-385

*"Whereas, adverse environmental exposure and the associated epigenetic **variations** could lead to prolonged negative effects on brain development and behavioral disadvantages."*

- 23) Minor – given the differing extent to which select groups wish to be considered to have ‘negative’ / less valued symptoms than those who are considered neurotypical, I suggest referring to polygenic scores (PGS) rather than polygenic *risk* scores (PRS) throughout. Either terms is widely used, but the latter suffers from being seen as possibly more negatively loaded.

Response: Thank you for your suggestions. We have changed all abbreviations of polygenic risk scores to PGS as suggested.

References

- 1 Blakemore, S. J. Adolescence and mental health. *Lancet* **393**, 2030-2031, doi:10.1016/S0140-6736(19)31013-X (2019).
- 2 Bethlehem, R. A. I. *et al.* Brain charts for the human lifespan. *Nature* **604**, 525-533, doi:10.1038/s41586-022-04554-y (2022).

REVIEWER COMMENTS

Reviewer #3 (Remarks to the Author):

No further comments - all of my comments were addressed.

Reviewer #4 (Remarks to the Author):

The authors have provided adequate responses to most of my points.

However, I would encourage the authors to check Supplementary Table 1. I do not think these can be the correct standardised loadings for the PCA. For example, knowing the proportion of variance explained (e.g. PC1 explains ~40%) and that this is calculated by summing the squares of the standardised loadings and dividing by the number of indicators, standardised loadings cannot be mainly ~0.04. To be clear, the standardised loading denotes the correlation (i.e. Pearson's r) between the indicator and the component/factor. I would expect the majority of loadings for a primary PC to be mainly $>.3$ or $.4$. Having an accurate indication of how the ROIs load on the factors is important for their subsequent interpretation, so it's important to provide this, and to get it right.

Point-by-Point Response to the Reviewers' Comments

Runye Shi, Shitong Xiang, Tianye Jia, Trevor W. Robbins, ..., Gunter Schumann, Xiaolei Lin*,
Barbara J. Sahakian*, Jianfeng Feng*, IMAGEN Consortium

Enclosed, please find the revised submission of the paper “Life-course investigation of structural neurodevelopment at the individual level”, for publication in Nature Communications. We are thankful to the comments from reviewer 4 and below we provide our response.

Reviewer #4 (Remarks to the Author):

The authors have provided adequate responses to most of my points.

However, I would encourage the authors to check Supplementary Table 1. I do not think these can be the correct standardised loadings for the PCA. For example, knowing the proportion of variance explained (e.g. PC1 explains ~40%) and that this is calculated by summing the squares of the standardised loadings and dividing by the number of indicators, standardised loadings cannot be mainly ~0.04. To be clear, the standardised loading denotes the correlation (i.e. Pearson's r) between the indicator and the component/factor. I would expect the majority of loadings for a primary PC to be mainly >.3 or .4. Having an accurate indication of how the ROIs load on the factors is important for their subsequent interpretation, so it's important to provide this, and to get it right.

Response: Thank you for your suggestion. We apologize for the confusion in understanding your previous comments. In our last point-to-point response letter, we mistakenly regarded the eigenvector matrix as factor loading and provided the 'eigenvector matrix' calculated by dividing factor loading by the corresponding eigenvalues. We have now provided the factor loading in the updated Supplementary Table 1, which referred to the correlation between the k-th principal component PC_k and the i-th indicator X_i as shown in the following formula:

$$\rho(PC_k, X_i) = \frac{\sqrt{\lambda_k} \alpha_{ik}}{\sqrt{\sigma_i}}$$

where λ_k is the k-th largest eigenvalue for the sample covariance matrix, α_k is the k-th eigenvectors and σ_i is the covariance of X_i (1 due to standardization). In this way, the sum of square factor loadings across all indicators on PC_k should be λ_k (the variance of PC_k)

Updated Supplementary Table 1. Rotation matrix, factor loading and proportion of variance explained by each PCA component.

	PC1	PC2	PC3	PC4	PC5	PC6	PC7	PC8	PC9	PC10	PC11	PC12	PC13	PC14	PC15
Cortical regions															
bankssts	0.75	0.12	-0.06	0.04	-0.05	0.21	-0.10	0.12	-0.18	0.08	-0.13	0.08	0.00	-0.06	-0.06
caudalanteriorcingulate	0.71	0.05	-0.04	-0.17	0.04	0.07	0.29	-0.19	0.16	0.07	-0.19	-0.08	0.08	0.14	-0.01
caudalmiddlefrontal	0.78	-0.25	-0.03	-0.27	0.04	0.01	0.15	0.04	0.08	0.07	0.11	-0.06	-0.05	-0.05	-0.01
cuneus	0.53	-0.29	0.19	0.37	0.28	-0.21	-0.19	-0.10	0.05	-0.11	-0.03	-0.07	0.00	-0.03	0.20
entorhinal	0.51	0.22	-0.24	0.09	0.10	0.19	-0.05	-0.09	0.27	-0.21	0.18	0.24	-0.07	-0.01	-0.07
fusiform	0.80	0.32	-0.17	0.12	0.10	0.06	-0.15	0.06	-0.03	-0.03	-0.08	0.02	0.06	-0.03	0.01
inferiorparietal	0.84	-0.14	0.00	0.02	-0.07	0.14	-0.04	0.13	-0.14	-0.05	-0.08	-0.01	-0.02	-0.12	-0.01

inferiortemporal	0.76	0.37	-0.25	0.03	0.04	0.06	-0.12	0.15	-0.04	-0.08	0.02	0.00	0.01	-0.02	0.09
isthmuscingulate	0.67	0.00	0.22	0.41	-0.06	-0.05	-0.10	-0.07	-0.01	0.08	-0.18	-0.01	0.04	0.03	-0.08
lateraloccipital	0.77	-0.13	0.02	0.16	0.14	-0.08	-0.15	0.12	-0.05	-0.05	-0.06	-0.01	0.03	-0.10	0.17
lateralorbitofrontal	0.84	0.15	-0.16	-0.09	-0.09	-0.01	0.01	-0.11	0.03	0.00	0.01	-0.07	-0.07	0.00	0.05
lingual	0.68	-0.03	0.07	0.37	0.29	-0.20	-0.13	-0.08	0.01	0.02	-0.11	-0.03	0.09	-0.11	-0.05
medialorbitofrontal	0.64	0.13	-0.06	-0.07	-0.26	-0.25	-0.01	-0.25	-0.06	-0.09	0.04	-0.04	0.02	0.02	0.03
middletemporal	0.80	0.32	-0.21	-0.04	0.03	0.08	-0.07	0.13	-0.10	-0.03	-0.08	0.00	0.02	-0.06	0.09
parahippocampal	0.49	0.37	-0.10	0.19	-0.05	0.12	-0.11	0.07	0.20	0.06	0.13	0.18	-0.16	0.04	-0.46
paracentral	0.62	-0.38	0.31	-0.09	-0.20	0.13	0.08	0.04	0.16	0.02	0.15	0.09	0.08	0.15	0.03
parsopercularis	0.76	-0.06	-0.07	-0.12	0.08	-0.10	0.01	-0.13	0.01	0.34	0.05	-0.07	-0.04	-0.04	0.01
parsorbitalis	0.75	0.04	-0.23	-0.18	0.04	-0.17	-0.04	-0.08	-0.04	0.02	0.08	-0.03	-0.09	-0.08	0.06
parstriangularis	0.74	-0.01	-0.14	-0.18	0.09	-0.17	-0.01	-0.09	0.01	0.34	0.10	-0.02	-0.06	-0.02	0.07
pericalcarine	0.39	-0.23	0.27	0.33	0.17	-0.36	-0.11	-0.07	0.19	-0.13	0.02	-0.21	-0.28	0.09	-0.23
postcentral	0.62	-0.41	0.29	-0.04	-0.22	0.05	0.02	0.07	-0.01	-0.19	0.08	-0.02	0.00	-0.01	0.00
posteriorcingulate	0.77	-0.05	0.24	0.04	-0.15	0.09	0.10	-0.11	0.15	0.15	-0.10	0.05	0.11	0.13	-0.09
precentral	0.82	-0.22	0.16	-0.15	-0.08	0.07	0.07	0.10	0.15	-0.06	0.11	0.02	0.00	0.01	-0.02
precuneus	0.83	-0.18	0.23	0.15	-0.07	0.15	-0.06	0.00	-0.01	0.01	-0.08	0.08	0.04	0.00	0.01
rostralanteriorcingulate	0.74	0.17	-0.14	-0.12	0.02	0.01	0.23	-0.19	0.08	-0.08	-0.23	-0.10	0.05	0.03	-0.05
rostralmiddlefrontal	0.84	-0.11	-0.22	-0.16	0.06	-0.15	0.02	-0.03	-0.03	0.10	0.08	-0.04	-0.08	-0.05	0.02
superiorfrontal	0.82	-0.27	-0.04	-0.28	-0.01	-0.08	0.10	0.00	0.06	0.08	0.12	-0.02	-0.05	-0.02	0.00
superiorparietal	0.73	-0.36	0.15	-0.03	-0.09	0.15	-0.08	0.14	-0.08	-0.17	0.00	0.04	-0.03	-0.08	0.05
superiortemporal	0.89	0.19	-0.08	-0.04	-0.01	0.05	-0.06	0.05	-0.04	0.09	-0.04	0.04	0.00	-0.01	0.01
supramarginal	0.79	-0.18	0.09	-0.05	-0.21	0.15	-0.08	0.13	-0.06	-0.15	-0.05	0.06	-0.01	-0.06	-0.04
frontalpole	0.29	-0.10	-0.09	-0.16	-0.17	-0.54	-0.07	-0.16	-0.42	-0.11	0.05	0.41	0.23	0.09	-0.21
temporalpole	0.51	0.24	-0.19	0.06	-0.01	0.09	-0.17	-0.28	0.09	-0.33	0.24	0.04	-0.02	0.35	0.25
transversetemporal	0.66	0.06	0.13	0.19	-0.02	0.09	-0.18	0.07	0.04	0.36	0.02	0.11	0.17	0.15	0.07
insula	0.60	0.03	-0.15	-0.11	0.06	0.12	0.20	-0.11	-0.04	-0.26	-0.37	-0.09	-0.05	-0.07	-0.08
Subcortical regions															
cerebellum_white_matter	-0.11	0.36	0.43	-0.24	-0.31	-0.14	-0.12	-0.14	0.09	0.01	-0.12	0.21	-0.47	-0.26	0.15
cerebellum_cortex	0.47	0.07	-0.04	-0.35	0.14	-0.25	-0.07	0.44	-0.04	-0.18	0.17	-0.26	0.03	0.05	-0.13
thalamus_proper	0.04	0.11	0.57	-0.39	0.35	-0.02	-0.10	0.01	-0.15	0.00	-0.21	0.19	-0.10	0.24	0.04
caudate	0.47	0.09	0.19	0.17	0.14	0.14	0.45	-0.12	-0.37	-0.10	0.05	-0.06	-0.13	0.14	-0.05
putamen	0.33	0.18	0.24	0.39	0.11	0.17	0.29	-0.10	-0.36	0.08	0.41	0.00	-0.12	-0.19	0.01
pallidum	0.00	0.27	0.46	-0.29	0.03	0.15	-0.21	-0.37	0.07	-0.12	0.15	-0.19	0.38	-0.36	-0.13
hippocampus	0.22	0.61	0.32	-0.02	-0.18	-0.20	0.08	0.23	-0.02	0.02	-0.04	-0.19	0.00	0.05	-0.11
amygdala	0.19	0.57	0.36	0.11	-0.31	-0.13	0.02	0.12	0.00	0.02	0.09	-0.20	0.09	0.17	0.19
accumbens_area	0.22	0.13	-0.02	0.32	-0.08	-0.38	0.52	0.20	0.30	-0.10	-0.03	0.27	0.17	-0.21	0.15
brain_stem	0.15	0.30	0.37	-0.30	0.60	0.03	0.12	0.11	0.13	-0.04	0.08	0.24	0.07	-0.01	0.03
Standard deviation	4.24	1.63	1.47	1.37	1.15	1.14	1.07	1.00	0.97	0.94	0.92	0.89	0.85	0.83	0.81
Proportion of variance	0.41	0.06	0.05	0.04	0.03	0.03	0.03	0.02	0.02	0.02	0.02	0.02	0.02	0.02	0.01
Cumulative proportion	0.41	0.47	0.52	0.56	0.59	0.62	0.65	0.67	0.69	0.71	0.73	0.75	0.76	0.78	0.79

REVIEWERS' COMMENTS

Reviewer #4 (Remarks to the Author):

Many thanks to the authors for now providing the requested information about the PCA analyses. Please make sure to provide clear information in the Results section text and the relevant Figure to indicate the proportion of variance explained. The first PC accounts for ~40% of the variation, and that the second and third PCs account only for 6% and 5%, respectively. To be clear, I do not think that this necessarily diminishes what was found, but I encourage the authors to consider and briefly reflect in the discussion what they make of the large differences in the amount of variance these PCs are capturing/reflecting about features of neurodevelopment.

I suggest that the authors update their interpretation of the family affirmation -> peak GMV by cg06064461. As mentioned previously, if the p-value does not survive multiple comparison correction, I do not think the unqualified statement that it was significant (e.g. Fig 4 e caption) is appropriate. This applies to the ~8 lines given over to interpretation of this non-significant result, and includes the erroneous statement that the magnitude of the mediation/effect size is somehow a direct function of sample size (lines 325-326).

Point-by-Point Response to the Reviewers' Comments

Runye Shi, Shitong Xiang, Tianye Jia, Trevor W. Robbins, ..., Gunter Schumann, Xiaolei Lin*, Barbara J. Sahakian*, Jianfeng Feng*, IMAGEN Consortium

Enclosed, please find the revised submission of the paper "Life-course investigation of structural neurodevelopment at the individual level", for publication in Nature Communications. We are thankful to the comments from reviewer 4 and below we provide our response.

Reviewer #4 (Remarks to the Author):

1. Many thanks to the authors for now providing the requested information about the PCA analyses. Please make sure to provide clear information in the Results section text and the relevant Figure to indicate the proportion of variance explained. The first PC accounts for ~40% of the variation, and that the second and third PCs account only for 6% and 5%, respectively. To be clear, I do not think that this necessarily diminishes what was found, but I encourage the authors to consider and briefly reflect in the discussion what they make of the large differences in the amount of variance these PCs are capturing/reflecting about features of neurodevelopment.

Response: Thanks for your suggestion. We have now added the proportion of variance explained by PC1 and PC2 in the Results section and provided a new subfigure in Supplementary Figure 2 with proportion of variance explained by each PC. For your second suggestion, as all brain regions have close connections with each other both anatomically and functionally, and follow very similar developmental patterns among healthy individuals, the large differences observed in the variance explained between PC1 and remaining PCs is actually not surprising. This result is also consistent with a previous study¹, where the authors did a PCA to all the regional GMVs for participants aged 7-20y and found that the top two PCs accounted for 51.9% and 2.9% of the total variance, respectively.

"The first and second PCs, which accounted for 41% and 6% of the variance, defined two combinations of GMV trajectories over the entire brain that were significantly associated with baseline total GMV (Supplementary Fig. 2b)."

Supplementary Fig. 2. The first two PCA components. Principal component analysis was used to define a low-dimensional representation of GMV developmental patterns. (a) Variance explained by each PC. The first 15 PCs, which explained 80% of the total variation were used in the downstream analysis...

2. I suggest that the authors update their interpretation of the family affirmation -> peak GMV by cg06064461. As mentioned previously, if the p-value does not survive multiple comparison correction, I do not think the unqualified statement that it was significant (e.g. Fig 4 e caption) is appropriate. This applies to the ~8 lines given over to interpretation of this non-significant result, and includes the erroneous statement that the magnitude of the mediation/effect size is somehow a direct function of sample size (lines 325-326)

Response: Thanks for your suggestion. We have revised the significance statement in a more proper way to avoid confusion.

Lines 319-328

*"However, given that only one site could be identified with differential methylation between Groups 1 and 2 **with relatively small sample size**, it should be noted that **higher level of family affirmation was associated with higher peak GMV through reduced cg06064461 methylation with an unadjusted p-value of 0.048** ($\beta = 0.005$, mediation proportion = 0.09, $P_{unadj} = 0.048$, $P_{adj} = 0.191$) (Fig. 4e). Family affirmation was defined as behaviors implemented by a parent to provide support or assistance to their children in diverse situations, demonstrating approval and affection and contributing to the parent-child relationship. These results indicated that environmental exposure could **potentially** contribute to disadvantaged neurodevelopment and neurocognition by inducing epigenetic differences of neurogenesis-related genes. **However, only a small mediation proportion was identified.**"*

*"Fig. 4e Mediation model was conducted to analyze the direct and indirect effect of family affirmation on peak GMV, with cg06064461 methylation as the mediator. Results showed that cg06064461 methylation mediated the relationship between family affirmation and peak GMV **with an unadjusted p-value of 0.048** ($\beta=0.005$, mediation proportion=9.26%, $*P_{unadj}=0.048$, $P_{adj}=0.191$)."*

References

- 1 Bray, S. Age-associated patterns in gray matter volume, cerebral perfusion and BOLD oscillations in children and adolescents. *Hum Brain Mapp* **38**, 2398-2407, doi:10.1002/hbm.23526 (2017).